# The Dawn of In Vivo Gene Editing Era: A Revolution in the Making

Sarfaraz K. Niazi

College of Pharmacy, University of Illinois, Chicago, IL 60612, USA; niazi@niazi.com

**Abstract:** Gene or genome editing (GE) revises, removes, or replaces a mutated gene at the DNA level; *it is a tool.* Gene therapy (GT) offsets mutations by introducing a "normal" version of the gene into the body while the diseased gene remains in the genome; *it is a medicine.* So far, no in vivo GE product has been approved, as opposed to 22 GT products approved by the FDA, and many more are under development. No GE product has been approved globally; however, critical regulatory agencies are encouraging their entry, as evidenced by the FDA issuing a guideline specific to GE products. The potential of GE in treating diseases far supersedes any other modality conceived in history. Still, it also presents unparalleled risks—from off-target impact, delivery consistency and long-term effects of gene-fixing leading to designer babies and species transformation that will keep the bar high for the approval of these products. These challenges will come to the light of resolution only after the FDA begins approving them and opening the door to a revolution in treating hundreds of untreatable diseases that will be tantamount to a revolution in the making. This article brings a perspective and a future analysis of GE to educate and motivate developers to expand GE products to fulfill the needs of patients.

**Keywords:** gene editing; CRISPR; gRNA; autoimmune disorders; nuclease; hereditary disorders





## 1. Introduction

The first reported instance of gene editing using "recombinant DNA" technology occurred in the early 1972 [1]. In 1972, Paul Berg, a biochemist at Stanford University, conducted an experiment where he successfully spliced together DNA from two different viruses. This experiment marked the first-time scientists could manipulate and edit genes in a controlled laboratory. While this experiment was not directly editing genes in the modern sense, it marked a critical milestone in genetic engineering. It set the stage for future advancements in gene editing technologies. Berg's pioneering work in gene editing was significant because it laid the foundation for developing modern gene-editing techniques. The term "gene editing", as we commonly understand it today, is more closely associated with technologies like CRISPR-Cas9, which emerged much later in the 2010s [2]. Since then, gene-editing technologies have advanced considerably, and in recent years, the CRISPR-Cas9 system has become the most widely used and versatile tool for gene editing.

Besides its role in treating diseases, GE has many applications in agriculture, biomaterials, biodiversity, biosensors, climate change, and even backtracking evolutionary cycles. The first reported gene editing in plants dates to the early 1980s, specifically in 1983, when researchers were able to introduce new genes into plant cells using recombinant DNA technology [3], inserting genes from the bacterium Agrobacterium tumefaciens into the tobacco plant cells, resulting in genetically modified tobacco plants.

The potential of gene editing as a therapeutic intervention for human disease extends across multiple diseases, such as cancer, blood disorders, genetic diseases, and viral diseases [4]. Gene editing technologies can correct disease-causing mutations in these diseases or equip cells with new functionalities to combat disease progression [5]. However, translating these technologies from bench to bedside has various hurdles, including ethical and

regulatory challenges and scientific and technological limitations [6]. This will limit their accessibility to a small fraction of patients who can afford them [7].

## 2. The Science of Gene Editing

The impact of GE is still not entirely predictable but realizing that inheritance patterns (recessive or dominant) and genetic etiology, many single-gene disorders that have been identified, lead to hundreds of incurable diseases, bring great promise to the utility of GE. Figure 1 shows the chromosomes related to disorders that have been confirmed, making these disorders an excellent target for gene editing. Single-gene diseases like sickle cell anemia, Tay-Sachs, Tay-Sachs, and hemochromatosis can be treated, as well as multiple gene mutation diseases, including several cancers.

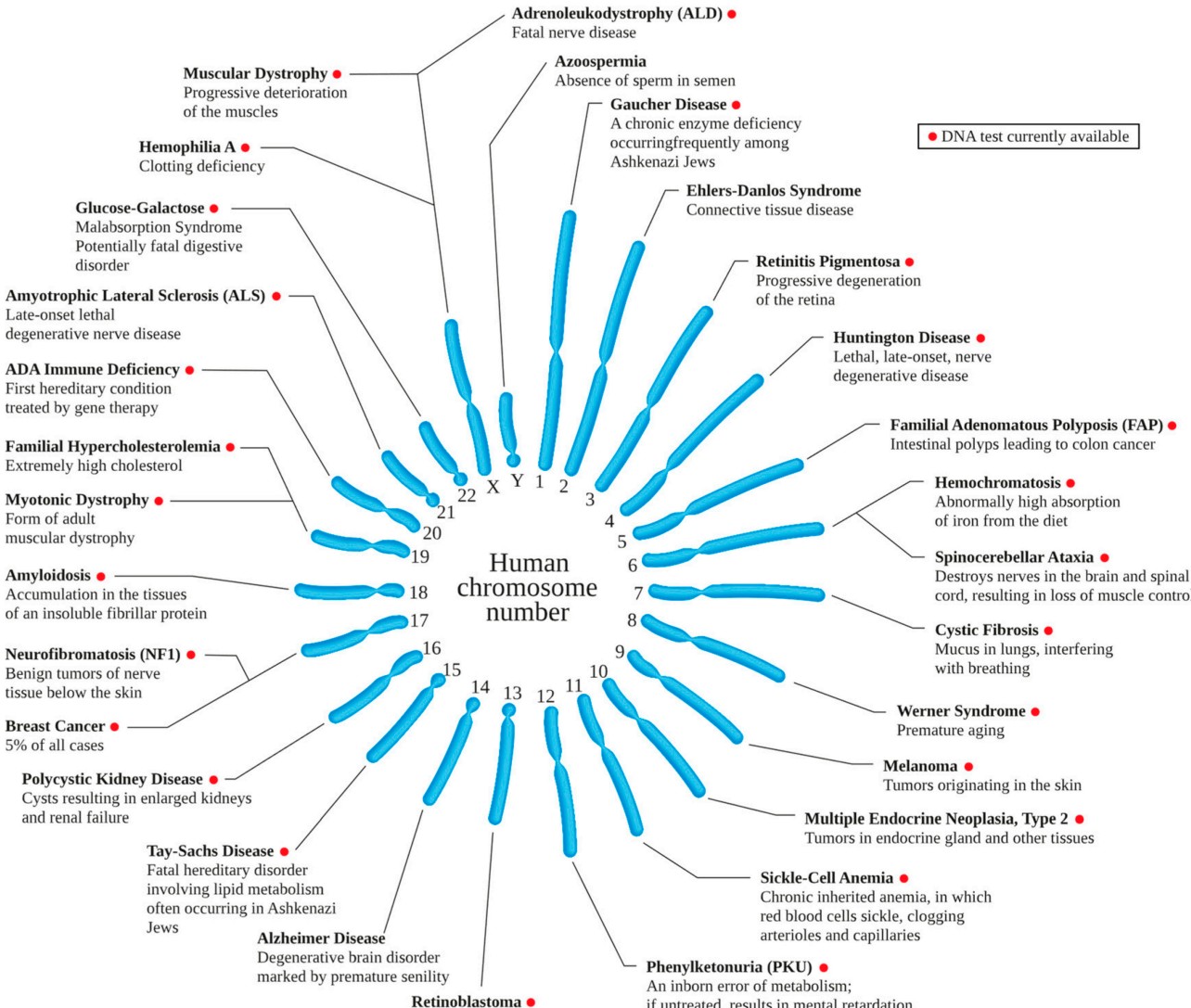

**Figure 1.** Genetic disorders. (In Wikipedia. https://en.wikipedia.org/wiki/Genetic_disorder, accessed on 26 February 2023).

Gene editing has emerged as a promising tool in cancer therapeutics. The technology allows precise modification of genes within an organism with a range of potential applications, such as correcting mutations in cancer cells or inducing lethal mutations in cancer cells. Using immunotherapy, gene editing can modify a patient's immune cells, specifically T-cells, to recognize and attack cancer cells; the cancer cells can be made more susceptible to chemotherapy or other treatments, enhancing the efficacy of existing treatments. Person-

alized cancer vaccines can be developed by understanding and editing genes responsible for antigen presentation. GE can directly target and correct gene mutations like TP53, a common mutation in many cancers [8]. Gene editing can also knock out genes in T-cells that express inhibitory receptors like PD-1, enhancing the immune response against cancer cells [9].

Many GE products are under development [10–14] as ex and in vivo products [15]. In vivo GE is the focus of GE tools, delivering the gene editing machinery into the body; ex vivo editing is considered at GT, which is already the scope of many approved products.

Ex vivo editing involves modifying cells outside the body and reintroducing them into the patient [16], such as using CRISPR/Cas9 to modify T cells ex vivo to express a novel T-cell receptor (TCR) that targets a cancer-specific antigen [17]. These GT approaches modify T cells, which are then re-infused into patients with advanced refractory cancers. They are also used in immuno-oncology treatments.

Figure 1 shows a most optimistic view of the future of GE, as for most genetic disorders, except in the case of Alzheimer's disease, Azospermia, and Ehlers-Danlos Syndrome, DNA testing has been developed; this allows identification of gene mutations, and thus designing GE tools. However, with the next generation sequencing (NGS), the DNA mutations will likely be recognized for most genetic disorders. What makes such exercise difficult is the heterogeneity of genes involved in many disorders that limits the focus of investigation. However, what is known today with certainty is large enough for decades of work on developing GE tools.

Chimeric antigen receptor (CAR) T cell therapies offer a significant promise, which involves modifying T cells to recognize and attack cancer cells. Two such treatments, Kymriah and Yescarta, have already received FDA approval, although they do not apply gene editing [18]. The ex vivo GE target cells include stem cells and immune cells that can be cultured and transplanted into the patient, such as SCID, ß-thalassemia, sickle cell anemia, and CAR-T therapy, including hematopoietic stem cells, T cells, and NK cells. These GT products introduce modified cells, leaving the mutated genes in place [19].

The more excellent utility of GE comes from an in vivo strategy that eliminates cell collection, isolation, expansion, editing, selection, and transplantation, making it more accessible and effective at targeting a single organ than the entire organism [20]. The GE is also ideally suited for precision or personalized therapies [21–23], primarily cancer treatment. In 2021, the approval of 17 personalized (individualized, precision) medicines represented approximately 35 percent of all newly approved therapeutic molecular entities [24].

The FDA has also established a secure, collaborative, high-performance computing platform that builds a community of experts around analyzing biological datasets to advance precision medicine [25].

*2.1. Next-Generation Sequencing*

The next-generation sequencing (NGS) technologies to discover new uncommon genetic illnesses have made individualized therapies more practical, revolutionizing the field of genomics and molecular biology. Its applications are developed in many areas of therapeutics. For example, NGS can identify tumor-specific mutations, allowing for the creation of personalized treatment plans based on the tumor's genetic makeup [26]. It can also help identify pathogens, track the source and spread of outbreaks more accurately [27], and detect congenital abnormalities in fetuses or newborns, ensuring timely interventions when necessary [28]. And for patients with rare and undiagnosed conditions, NGS can provide answers and guide treatment plans [29].

*2.2. Personalized Medicine*

Interestingly, gene editing technologies have opened new therapeutic possibilities and brought about a new era of personalized medicine. Therapies can be designed according to the genetic profile of individual patients, thereby improving therapeutic outcomes while minimizing potential adverse effects [30]. For example, gene editing technologies could

engineer immune cells to recognize and kill cancer cells more effectively in the context of cancer [17]. Despite these exciting prospects, there are legitimate concerns surrounding gene editing technologies, particularly when it comes to alterations that can be inherited by future generations (germline gene editing) [31].

*2.3. Targeting*

Gene editing treatments should target multiple, conserved, and functionally essential sites on genomes and use numerous gene editing events to prevent the development of resistance because gene editing encounters biological resistance immunologically or by selecting cells with a resistant target region. The current understanding includes the realization that while a single gene causes these disorders, different mutations can cause the same disease with a range of severity. In addition, several phenotypes can occur from the same mutation, brought on by variances in the patient's surroundings and other genetic variations. For instance, additional genes have been shown to modify the cystic fibrosis phenotype in infants with the same mutation in the cystic fibrosis transmembrane conductance regulator (CFTR); for some diseases like galactosemia, mutations in various genes can produce similar manifestations [32].

Table 1 lists the main identified gene editing applications that can significantly impact human health.

**Table 1.** Potential applications of gene editing to treat human diseases in alphabetical order.

| Condition or Disease | GE Applications |
|---|---|
| Aging | Genetic modification of senescent cells has been proposed to counteract aging [33]. |
| Alkaptonuria | A metabolic disorder that leads to a buildup of homogentisic acid, causing various symptoms, including dark-colored urine. Gene editing could correct the HGD gene, which causes this condition [34]. |
| Alpha-1 Antitrypsin Deficiency | This genetic disorder can lead to lung and liver disease. Gene editing technologies can potentially correct the faulty SERPINA1 gene that causes it [35]. |
| Alport Syndrome | A genetic disorder characterized by kidney disease, hearing loss, and eye abnormalities. Gene editing could correct the COL4A3, COL4A4, or COL4A5 genes, which cause this condition [36]. |
| Amyotrophic Lateral Sclerosis (ALS) | A group of neurological diseases mainly involving the nerve cells (neurons) responsible for controlling voluntary muscle movement. Gene editing might correct the SOD1, TARDBP, FUS, or C9orf72 genes, which can cause this condition [37]. |
| Angioedema | A rare genetic disorder characterized by recurrent episodes of severe swelling. Gene editing might correct the SERPING1 gene, which causes this condition [38]. |
| Antimicrobial Resistance | Gene editing can potentially counteract the growing problem of antibiotic resistance by directly targeting and killing antibiotic-resistant bacteria or by making the bacteria sensitive to antibiotics again [39]. |
| Atherosclerosis | Gene editing can potentially treat atherosclerosis, a disease where plaque builds up inside the arteries. Using gene editing techniques to modify the PCSK9 gene, which controls LDL cholesterol levels, could potentially lower the risk of atherosclerosis [40]. |
| Autoimmune Diseases | With autoimmune diseases like rheumatoid arthritis and lupus, gene editing could modify immune cells and prevent them from attacking the body's tissues [41]. |
| Bardet-Biedl Syndrome | A disorder that affects many parts of the body and can cause obesity, loss of vision, kidney abnormalities, and extra fingers or toes. Gene editing could correct any 21 genes that cause this condition, such as BBS1, BBS2, or BBS10 [42]. |
| Barth Syndrome | A genetic disorder characterized by muscle weakness, delayed growth, and sometimes intellectual disability. Gene editing could correct the TAZ gene, which causes this condition [43]. |
| Beta Thalassemia | Gene editing techniques have been used in attempts to treat beta-thalassemia, a blood disorder that reduces the production of hemoglobin. Researchers have manipulated the BCL11A gene to enhance the production of fetal hemoglobin as a workaround [44]. |
| Blindness | Researchers have used gene editing to restore sight in blind mice, which could eventually lead to treatments for certain forms of inherited blindness in humans [45]. |

**Table 1.** *Cont.*

| Condition or Disease | GE Applications |
|---|---|
| Brain Function | Gene editing has been used in neuroscience to understand the function of different genes in the brain, which could eventually help us treat or even cure neurological disorders [46]. |
| Canavan Disease | A progressive, fatal neurological disorder that begins in infancy. Gene editing could correct the ASPA gene, which causes this condition [47]. |
| Cancer | CRISPR has been used to develop novel cancer therapies, such as genetically modifying a patient's immune cells to target and fight cancer cells [48]. |
| Chronic Granulomatous Disease | Gene editing has been used to correct the genetic mutations that cause chronic granulomatous disease, a disorder that affects the immune system [49]. |
| Cystic Fibrosis | Gene editing technologies could correct the CFTR gene mutation that leads to cystic fibrosis, a disease affecting the lungs and digestive system [50]. |
| Cystinosis | A genetic disorder characterized by an accumulation of the amino acid cystine within cells, leading to various symptoms and complications. Gene editing technologies are being explored to correct the CTNS gene, which causes this condition [51]. |
| Diabetes | In Type 1 diabetes, the immune system destroys insulin-producing cells. Researchers are exploring gene editing as a potential approach to protect these cells from the immune system or to create new insulin-producing cells [52]. |
| Duchenne Muscular Dystrophy | A severe type of muscular dystrophy. Gene editing has shown promise in correcting the gene mutation that causes this condition [53]. |
| Dystonia | A movement disorder in which a person's muscles contract uncontrollably. Gene editing could correct any of the 20+ known genes that can cause this condition, such as TOR1A, THAP1, or GNAL [54]. |
| Epidermolysis Bullosa | Gene editing technologies can potentially correct the genetic mutations that cause epidermolysis bullosa, a group of genetic conditions that cause the skin to be very fragile and to blister easily [55]. |
| Familial Exudative Vitreoretinopathy | A hereditary disorder that can cause progressive vision loss. Gene editing could correct the FZD4, LRP5, TSPAN12, NDP, or ZNF408 genes, which cause this condition [56]. |
| Fanconi Anemia | A rare genetic disease resulting in bone marrow failure. Gene editing could correct the FANCC gene, one of the known genes that causes this condition when mutated [57]. |
| Fragile X Syndrome | A genetic disorder causing intellectual disability, behavioral and learning challenges, and various physical characteristics. Gene editing has been proposed as a potential way to correct the FMR1 gene that causes this condition [58]. |
| Gaucher Disease | A genetic disorder that affects the body's ability to break down fats. Gene editing technologies are being explored to correct the GBA gene mutations that cause this condition [59]. |
| Glycogen Storage Disease Type Ia | A metabolic disorder caused by the deficiency of glucose-6-phosphatase, the enzyme necessary for the final step of gluconeogenesis and glycogenolysis. Gene editing technologies are being explored to correct the G6PC gene mutations that cause this condition [60]. |
| Gorlin Syndrome | A genetic condition affecting many body parts increases the risk of developing various cancerous and noncancerous tumors. Gene editing could correct the PTCH1 gene, which causes this condition [61]. |
| Hemophilia A and B | A rare bleeding disorder in which a person lacks or has low levels of specific proteins is called "clotting factors". Gene editing might correct the F8 or F9 genes, which cause Hemophilia A and B, respectively [62]. |
| Hemorrhagic Telangiectasia | A genetic disorder of blood vessel formation causes multiple direct connections between arteries and veins. Gene editing could correct the ENG, ACVRL1, or SMAD4 gene, which can cause this condition [63]. |
| HIV/AIDS | Gene editing technology has been used to eradicate HIV from infected cells. This is achieved by targeting the viral DNA integrated into the host genome [64]. |
| Huntington's Disease | This inherited disease causes the progressive breakdown of nerve cells in the brain. Gene editing could potentially correct or deactivate the gene that causes Huntington's disease [65]. |
| Hypertrophic Cardiomyopathy | A disease in which the heart muscle becomes abnormally thick, making it harder for the heart to pump blood. Gene editing might correct the MYH7 gene, which causes this condition [66]. |

**Table 1.** *Cont.*

| Condition or Disease | GE Applications |
|---|---|
| Hypophosphatasia | A metabolic disease that disrupts mineralization, processes in which minerals such as calcium and phosphorus are deposited in developing bones and teeth. Gene editing could correct the ALPL gene, which causes this condition [67]. |
| Immunodeficiencies | Gene editing may provide treatments for primary immunodeficiencies, like severe combined immunodeficiency (SCID), where gene alterations affect the immune system's development and function [68]. |
| Infertility | Gene editing technology may be utilized to treat genetic disorders that cause infertility, offering hope to many individuals and couples wishing to have children [69]. |
| Joubert Syndrome | A genetic disorder characterized by decreased muscle tone, difficulties with coordination, abnormal eye movements, and breathing problems. Gene editing could correct any of the 35 genes that cause this condition, such as AHI1, CEP290, or TMEM67 [70]. |
| Juvenile Polyposis Syndrome | A genetic condition characterized by multiple polyps in the gastrointestinal tract. Gene editing could correct the BMPR1A or SMAD4 genes, which cause this condition [71]. |
| Leber Congenital Amaurosis | An eye disorder that primarily affects the retina. Gene editing could correct any of the 14 known genes that can cause this condition, such as GUCY2D, RPE65, or CEP290 [72]. |
| Leukemia | Specific genetic mutations cause certain forms of leukemia. Gene editing could potentially correct these mutations, leading to improved treatment outcomes [73]. |
| Li-Fraumeni Syndrome | A rare, hereditary disorder that significantly increases the risk of developing several types of cancer, particularly in young adults and children. Gene editing could correct the TP53 gene, which causes this condition [74]. |
| Long QT Syndrome | A disorder of the heart's electrical activity can cause sudden, uncontrollable, and irregular heartbeats (arrhythmias), which may lead to premature death. Gene editing could correct any of the 17 known genes that can cause this condition, such as KCNQ1, KCNH2, or SCN5A [75]. |
| Lynch Syndrome | A genetic condition that increases the risk of many types of cancer, particularly colorectal cancers. Gene editing could correct the MSH2, MLH1, MSH6, or PMS2 genes, which cause this condition [76]. |
| Marfan Syndrome | A genetic disorder affecting the body's connective tissue. Gene editing has been proposed to correct the faulty gene that causes this syndrome [77]. |
| Mitochondrial Diseases | Mitochondrial diseases often result from mutations in the mitochondrial DNA. Scientists have used gene editing techniques to selectively eliminate mutated mitochondrial DNA and prevent these diseases [78]. |
| Mucopolysaccharidosis | A group of metabolic disorders caused by the absence or malfunctioning of lysosomal enzymes needed to break down molecules called glycosaminoglycans. Gene editing could correct any of the 11 known genes that cause these conditions [79]. |
| Multiple System Atrophy | A rare neurodegenerative disorder characterized by autonomic dysfunction, parkinsonism, and ataxia. Gene editing could investigate and possibly correct the underlying genetic contributors to this condition, which are not yet fully understood [80]. |
| Muscular Dystrophy | Scientists have used gene editing to correct the mutation that causes Duchenne muscular dystrophy in animal models, and clinical trials are in the works [53]. |
| Neurodegenerative Disorders | Gene editing can be employed to study and potentially treat neurodegenerative disorders like Parkinson's, Alzheimer's, and Huntington's disease by targeting the specific genes involved in these conditions [81]. |
| Neurofibromatosis | Genetic disorders that cause tumors to form on nerve tissue. Gene editing could correct the NF1 or NF2 genes that cause these conditions [82]. |
| Niemann-Pick Disease | A group of severe inherited metabolic disorders in which sphingomyelin accumulates in cell lysosomes. Gene editing could correct the SMPD1 gene, which causes types A and B of this disease [83]. |
| Oculocutaneous Albinism | A group of conditions that affect the coloring (pigmentation) of the skin, hair, and eyes. Gene editing could correct the OCA2 gene, causing some forms of this condition [84]. |
| Osteogenesis Imperfecta | This group of genetic disorders mainly affects the bones, resulting in bones that break easily. Gene editing could correct or compensate for the faulty genes causing these conditions [85]. |

**Table 1.** *Cont.*

| Condition or Disease | GE Applications |
|---|---|
| Osteopetrosis | A group of rare, genetic bone disorders that result in the bone being overly dense. Gene editing could correct the TCIRG1, CLCN7, or SNX10 genes, which cause this condition [86]. |
| Pantothenate Kinase-Associated Neurodegeneration (PKAN) | A type of neurodegeneration with brain iron accumulation. Gene editing could correct the PANK2 gene, which causes this condition [87]. |
| Paraganglioma and Pheochromocytoma | Rare neuroendocrine tumors originate in the adrenal glands or near specific nerves and blood vessels. Gene editing could correct the SDHA, SDHB, SDHC, SDHD, SDHAF2, VHL, RET, NF1, TMEM127, or MAX genes, which can cause these conditions [88]. |
| Peutz-Jeghers Syndrome | A genetic condition characterized by the development of noncancerous growths called hamartomatous polyps in the gastrointestinal tract and a significantly increased risk of developing certain types of cancer. Gene editing could correct the STK11 gene, which causes this condition [89]. |
| Polycystic Kidney Disease | A genetic disorder characterized by the growth of numerous cysts in the kidneys. Gene editing might correct the PKD1 or PKD2 genes, which cause this condition [90]. |
| Pompe Disease | A metabolic disorder is caused by the buildup of a complex sugar called glycogen within cells. Gene editing could correct the GAA gene, which causes this condition [91]. |
| Prader-Willi Syndrome | This complex genetic condition affects many body parts, causing weak muscle tone, feeding difficulties, poor growth, and delayed development. Using gene editing to reactivate the silenced paternal copy of the genes could provide a cure [92]. |
| Progeria | Gene editing technology has shown promise in treating Progeria (also known as Hutchinson-Gilford Progeria Syndrome), a rare, fatal genetic disorder characterized by an appearance of accelerated aging in children. Gene editing can potentially correct the mutation in the LMNA gene associated with this disease [93]. |
| Retinal Diseases | Gene editing holds promise in treating inherited retinal diseases. Scientists have successfully used gene editing techniques to correct a mutation causing Leber congenital amaurosis, inherited blindness, breakdown, and loss of cells in the retina. Gene editing might correct any of the 60+ known genes that can cause this condition, such as RHO, RPGR, or USH2A [94]. |
| Rett Syndrome | A rare genetic disorder causing severe cognitive and physical impairments. Gene editing could potentially reactivate the silenced MECP2 gene that causes Rett syndrome [95]. |
| Sanfilippo Syndrome | A type of Mucopolysaccharidosis, Sanfilippo syndrome is characterized by the body's inability to break down certain sugars properly. Gene editing technologies are being developed to correct the SGSH gene, which causes this condition [96]. |
| Sickle Cell Disease | Gene editing has shown promise in correcting the genetic mutation responsible for sickle cell disease, which causes misshapen red blood cells [97]. |
| Spastic Paraplegia | A group of inherited disorders characterized by progressive weakness and stiffness of the legs. Gene editing could correct the SPG11 gene, which causes one of the more common types of this disease [98]. |
| Spinocerebellar Ataxias | These genetic diseases are characterized by degenerative changes in the part of the brain related to movement control. Gene editing techniques have been applied in experimental models to correct the associated genetic mutations [99]. |
| Tuberous Sclerosis Complex | A genetic disorder characterized by the growth of numerous noncancerous (benign) tumors in many body parts. Gene editing could correct the TSC1 or TSC2 genes that cause this condition [100]. |
| Tyrosinemia Type 1 | A rare genetic disorder characterized by multistep disruptions that break down the amino acid tyrosine. Gene editing could correct the FAH gene mutations causing this condition [101]. |
| Waardenburg Syndrome | A group of genetic conditions that can cause hearing loss and changes in coloring (pigmentation) of the hair, skin, and eyes. Gene editing could correct the PAX3 or MITF genes, which cause this condition [102]. |
| Werner Syndrome | A disorder characterized by the premature appearance of features associated with normal aging. Gene editing could correct the WRN gene, which causes this condition [103]. |
| Wilson Disease | A condition where copper builds up in the body, potentially leading to life-threatening organ damage. Gene editing might correct the ATP7B gene mutations causing Wilson's disease [104]. |
| Wolfram Syndrome | A genetic disorder characterized by diabetes mellitus and progressive vision loss. Gene editing could correct the WFS1 or CISD2 genes, which cause this condition [105]. |

**Table 1.** *Cont.*

| Condition or Disease | GE Applications |
|---|---|
| X-Linked Agammaglobulinemia | Gene editing can potentially correct mutations in the BTK gene, which cause X-linked agammaglobulinemia, an immune system disorder that leaves the body prone to infections [106]. |
| Xenotransplantation | Researchers have used gene editing to remove retroviruses from pig genomes, bringing us one step closer to pig-to-human organ transplants [107]. |
| Zellweger Spectrum Disorder | A group of conditions that can affect many body parts. Gene editing could correct any 12 PEX genes known to cause these conditions [108]. |

## 3. Current Status

Genetic mutations trigger human evolution and bring thousands of diseases. Until gene manipulation technologies arrived, treating the ailments of about a billion patients suffering from mutated, disease-causing genes was impossible. GE technology is preceded by GT products that fall within the exact definition, and understanding the nature of GT products is essential for GE development. The entry of GT products began in 2017 using an AAV-delivered ZFN [109].

The FDA has approved 22 GT products [110] (Table 2), and the EMA [111] has approved 13 products (labeled as Advanced Therapy Medicinal Products) (ATMPs) that the FDA also approves; neither agency has approved any GE product, though a large number are under development [112,113]. Clinicaltrials.gov lists 65 gene editing interventional trials, of which 13 have been completed, including two early Phase 1, 41 Phase 1 (41), 24 Phase 2, six Phase 3, no Phase 4, and ten other studies [114].

**Table 2.** FDA-approve Gene Therapy (GT) products [current as of August 2023].

| Product | Developer | Indication |
|---|---|---|
| ABECMA (idecabtagene vicleucel) | Celgene Corporation | Adult patients with relapsed or refractory multiple myeloma after four or more prior lines of therapy, including an immunomodulatory agent, a proteasome inhibitor, and an anti-CD38 monoclonal antibody. |
| ADSTILADRIN | Ferring Pharmaceuticals A/S | Adult patients with high-risk Bacillus Calmette-Guérin (BCG)-unresponsive non-muscle invasive bladder cancer (NMIBC) with carcinoma in situ (CIS) with or without papillary tumors |
| BREYANZI | Juno Therapeutics, Inc. | Adult patients with large B-cell lymphoma (LBCL), including diffuse large B-cell lymphoma (DLBCL) not otherwise specified (including DLBCL arising from indolent lymphoma), high-grade B-cell lymphoma, primary mediastinal large B-cell lymphoma, and follicular lymphoma grade 3B, who have: Refractory disease to first-line chemoimmunotherapy or relapse within 12 months of first-line chemoimmunotherapy; or Refractory disease to first-line chemoimmunotherapy or relapse after first-line chemoimmunotherapy and are not eligible for hematopoietic stem cell transplantation (HSCT) due to comorbidities or age; or Relapsed or refractory disease after two or more lines of systemic therapy. |
| CARVYKTI (ciltacabtagene autoleucel) | Janssen Biotech, Inc. | Adult patients with relapsed or refractory multiple myeloma after four or more prior lines of therapy, including a proteasome inhibitor, an immunomodulatory agent, and an anti-CD38 monoclonal antibody. |
| ELEVIDYS delandistrogene moxeparvovec | Sarepta Therapeutics, Inc. | Ambulatory pediatric patients aged 4 through 5 years with Duchenne muscular dystrophy (DMD) with a confirmed mutation in the DMD gene. |
| GINTUIT (Allogeneic Cultured Keratinocytes and Fibroblasts in Bovine Collagen) | Organogenesis Incorporated | It is an allogeneic cellularized scaffold product indicated for topical (non-submerged) application to a surgically created vascular wound bed in mucogingival conditions in adults. |
| HEMGENIX | CSL Behring LLC | HEMGENIX is an adeno-associated virus vector-based gene therapy indicated for adults with Hemophilia B (congenital Factor IX deficiency) who use Factor IX prophylaxis therapy, have current or historical life-threatening hemorrhage, or have repeated, spontaneous severe bleeding episodes. |

**Table 2.** *Cont.*

| Product | Developer | Indication |
|---|---|---|
| IMLYGIC (talimogene laherparepvec) | BioVex, Inc. | For the local unresectable cutaneous, subcutaneous, and nodal lesions in patients with recurrent melanoma after initial surgery. |
| KYMRIAH (tisagenlecleucel) | Novartis Pharmaceuticals Corporation | KYMRIAH is a CD19-directed genetically modified autologous T-cell immunotherapy indicated for adult patients with relapsed or refractory follicular lymphoma after two or more lines of therapy |
| LANTIDRA (donislecel) | CellTrans Inc. | Adults with Type 1 diabetes who cannot approach target hba1c because of repeated episodes of severe hypoglycemia despite intensive diabetes management and education. |
| LAVIV (Azficel-T) | Fibrocell Technologies | Improvement of the appearance of moderate to severe nasolabial fold wrinkles in adults. |
| LUXTURNA | Spark Therapeutics, Inc. | Patients with confirmed biallelic RPE65 mutation-associated retinal dystrophy. |
| MACI (Autologous Cultured Chondrocytes on a Porcine Collagen Membrane) | Vericel Corp. | Repair single or multiple symptomatic, full-thickness cartilage defects of the knee with or without bone involvement in adults. MACI is an autologous cellularized scaffold product. |
| OMISIRGE (omidubicel-onlv) | Gamida Cell Ltd. | Adults and pediatric patients 12 years and older with hematologic malignancies are planned for umbilical cord blood transplantation following myeloablative conditioning to reduce the time to neutrophil recovery and the incidence of infection. |
| PROVENGE (sipuleucel-T) | Dendreon Corp. | Asymptomatic or minimally symptomatic metastatic castrate-resistant (hormone refractory) prostate cancer. |
| RETHYMIC | Enzyvant Therapeutics GmbH | For immune reconstitution in pediatric patients with congenital athymia. |
| ROCTAVIAN (valoctocogene roxaparvovec-rvox) | BioMarin Pharmaceutical Inc | Adults with severe hemophilia A (congenital factor VIII deficiency with factor VIII activity <1 IU/dL) without pre-existing antibodies to adeno-associated virus serotype five detected by an FDA-approved test. |
| SKYSONA (elivaldogene autotemcel) | Bluebird bio, Inc. | To slow the progression of neurologic dysfunction in boys 4–17 years of age with early, active cerebral adrenoleukodystrophy (CALD). |
| STRATAGRAFT | Stratatech Corporation | Adults with thermal burns containing intact dermal elements for which surgical intervention is clinically indicated (deep partial-thickness burns). |
| TECARTUS (brexucabtagene autoleucel) | Kite Pharma, Inc. | Adult patients with relapsed or refractory mantle cell lymphoma (MCL). New Indication for this supplement: Adult patients with relapsed or refractory (r/r) B-cell precursor acute lymphoblastic leukemia (ALL) |
| VYJUVEK | Krystal Biotech, Inc. | Wounds in patients six months of age and older with dystrophic epidermolysis bullosa with mutation(s) in the *collagen type VII alpha one chain (COL7A1) gene* |
| YESCARTA (axicabtagene ciloleucel) | Kite Pharma, Incorporated | Adult patients with large B-cell lymphoma refractory to first-line chemoimmunotherapy or relapse within 12 months of first-line chemoimmunotherapy. Axicabtagene ciloleucel is not indicated in patients with primary central nervous system lymphoma. |
| ZOLGENSMA (onasemnogene abeparvovec-xioi) | Novartis Gene Therapies, Inc. | Adult and pediatric patients with ß-thalassemia who require regular red blood cell (RBC) transfusions |
| ZYNTEGLO (betibeglogene autotemcel) | Bluebird Bio, Inc. | Spinal muscular atrophy (type I) |

The first FDA-approved GT product, Kymriah (tisagenlecleucel), an antigen receptor T cell that is chimeric (CAR-T) [115], costs about half a million dollars per dose; the most recent gene therapy product, Hemgenix, a hemophilia treatment, costs $3.5 million per dose [116]. This high cost comes from the amortization of development costs of about $1–5 billion per product [117] distributed over a smaller number of patients. Future GE products are expected to have a high cost, opening the debates about the affordability of these products.

**4. GE Tools**

GE involves homologous recombination (HR), wherein nucleotide sequences between two DNA molecules that are similar or identical are switched; this has long been thought

to be a treatment for human genetic illnesses. However, its effectiveness is enhanced by causing DNA double-strand breaks (DSBs) using nucleases, a significant advancement in GE technology.

### 4.1. Nuclease Mediated

Nucleases are enzymes involved in the hydrolysis of the phosphodiester bonds that link nucleotides in nucleic acid chains. These enzymes are crucial for many biological processes, including DNA replication, recombination, repair, and RNA processing.

Endonucleases cleave the phosphodiester bonds within the nucleic acid chain, cutting the chain into smaller fragments. The most famous is the restriction endonuclease, which cuts DNA at specific sequences. Examples include EcoRI and HindIII [118]. Exonucleases, which remove nucleotides from the ends of nucleic acid chains. They may act at the 3′ end or 5′ end. Examples include ExoI and ExoIII [119]. DNase (Deoxyribonucleases) specifically acts on DNA. They can be endo or exo-nucleases such as DNase I or II [120]. RNase (Ribonucleases) specifically acts on RNA. They can also be either endo or exo-nucleases such as RNase A RNase H [121]. Restriction-modification systems found in bacteria work in tandem with a modification enzyme to recognize and cut foreign DNA while leaving the host DNA unharmed, such as EcoRI methylase with EcoRI [122]. CRISPR-associated nucleases are part of the CRISPR adaptive immune system in bacteria and archaea, such as Cas9 and Cas12 [123]. Homing Endonucleases introduce double-strand breaks at specific locations within genomes, such as I-SceI [124]. DNA Repair Enzymes specialize in repair pathways, such as mismatch repair, or base-excision repair, such as XPF-ERCC1 FEN1 [125].

Several engineered nucleases have been developed for targeted genome editing. Much of today's basic science supporting GE tools is built on restriction enzymes to cut DNA, discovered in 1968 [126]. Meganucleases (MNs), Zinc Finger Nucleases (ZFNs), Transcription Activator-like Effector Nucleases (TALENs), and Clustered Regularly Interspaced Short Palindromic Repeats-linked proteins (CRISPR-Cas), the sequence-specific nucleases. These nucleases are the most frequently used methods to correct genetic mutations.

ZFN are artificial restriction enzymes generated by fusing a zinc finger DNA-binding domain to a DNA-cleavage domain. This allows for the specific targeting of desired DNA sequences, such as Zif268 [127]. Meganucleases are characterized by a large recognition site (14–40 base pairs). Due to their high specificity, they are less likely to cut non-target sites in the genome. These are sometimes naturally occurring, like I-SceI, but can also be engineered [128]. Transcription Activator-Like Effector Nucleases (TALENs) are like ZFNs but use a different DNA-binding domain of Transcription Activator-Like Effectors (TALEs), proteins secreted by Xanthomonas bacteria when they infect plants. The TALE domains can be engineered to bind almost any desired DNA sequence, such as AvrBs3 [129]. These engineered nucleases have revolutionized the field of genome editing, offering unprecedented control over the modification of genetic material. They are used in various applications, from basic research to clinical gene therapy. Like with CRISPR-associated nucleases, these engineered nucleases create specific double-stranded breaks in the DNA at the location specified by their DNA-binding domains. Cellular machinery then repairs these breaks, and in doing so, changes can be introduced, allowing for the deletion, insertion, or replacement of specific DNA sequences.

In theory, the GE tools, the engineered nucleases, substitute one or more bases in any desired gene at any specific position. These technologies used programmable endonucleases and specific DNA target recognition sequences to create DNA double-strand breaks (DSBs), which led to gene replacements, insertions, deletions, and nucleotide substitutions. The repair of DSBs is required to preserve genetic material, but misrepair of DSBs can cause local sequence alteration or gross chromosomal rearrangements. The two main mechanisms to repair DSBs are classical nonhomologous end joining (C-NHEJ) and homologous recombination (HR). HR is generally considered an error-free mechanism because the homologous sister chromosome templates repair in S or G2 phase cells. C-NHEJ involves

the direct ligation of DNA ends and can occur with high fidelity or is associated with small alterations at the junctions [130].

Figure 2 shows (A) the meganuclease, I-SceI, binding its DNA target, where the catalytic domain, which determines DNA sequence specificity, is shown in red. (B) A ZFN dimer is illustrated bound to DNA. ZFN targets are bound by two zinc-finger DNA binding domains (dark blue) separated by a 5–7-bp spacer sequence. FokI cleavage occurs within the spacer. Each zinc finger typically recognizes 3 bp. (C) Depicted is a TALEN dimer bound to DNA. The DNA binding domains are dark blue. A 15–20-bp spacer sequence typically separates the two TALEN target sites. Like ZFNs, the TAL effector repeat arrays are fused to FokI. Each TAL effector motif recognizes one base. (D) The CRISPR/Cas9 system recognizes DNA through base pairing between DNA sequences at the target site and a CRISPR-based guide RNA (gRNA). Cas9 has two nuclease domains (shown by red arrowheads) that each cleave one strand of double-stranded DNA [131].

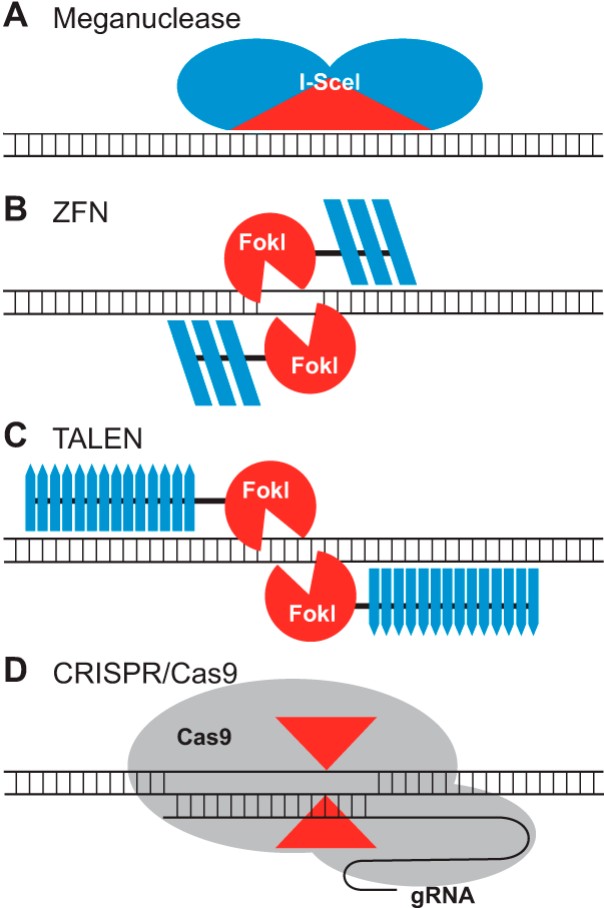

**Figure 2.** The four classes of nucleases that can provide the gene editing function [131], Reproduced under Creative Commons Attribution License. (**A**) Meganuclease; (**B**) ZFD; (**C**) TALEN; (**D**) CRISPR/Cas9.

Figure 3 shows how targeted genomes are engineered by nonhomologous end-joining (NHEJ) or homologous recombination using sequence-specific nucleases. (A) The NHEJ-mediated repair can result in small deletions or insertions at the target sites that can disrupt gene function (knockouts, left). DNA fragments can be inserted via NHEJ-mediated ligation to create targeted insertions (knock-ins, right). (B) When SSNs make two cuts, NHEJ-mediated repair can result in deletions or inversions of large genomic regions (left), targeted gene deletions, or chromosomal translocations (right). (C) HR-mediated repair, involving a homologous DNA template, leads to gene replacement or insertion.

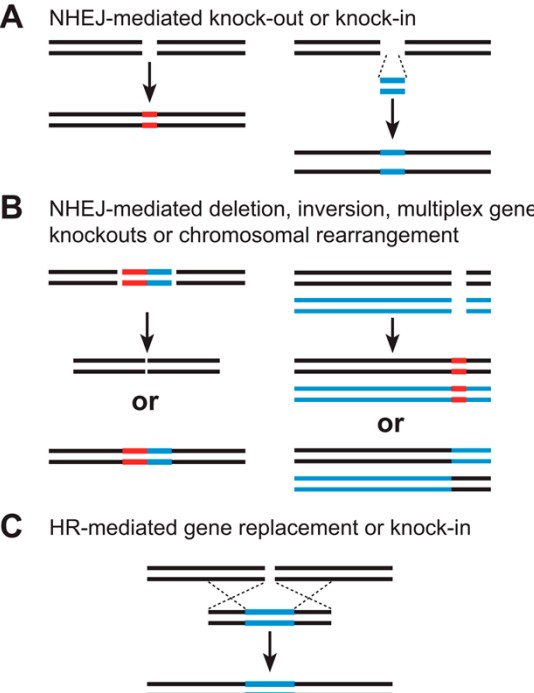

**Figure 3.** Targeted genomes are engineered by nonhomologous end-joining or homologous recombination using sequence-specific nucleases [131], Reproduced under Creative Commons Attribution License. (**A**) NHEJ-mediated knock-out or knock-in (**B**) NHEJ-mediated deletion, inversion, multiplex gene knockouts or chromosomal rearrangement (**C**) HR-mediated gene replacement or knock-in.

Table 3 provides a comparative analysis of the commonly used nucleases with utility, advantages, and disadvantages.

**Table 3.** Comparison of key nuclease-mediated GE technologies currently in use [132]. + = applicability.

| Attribute | Meganucleases | Zinc Finger Nucleases | TALENs | CRISPR/Cas9 |
|---|---|---|---|---|
| Enzyme | Endonuclease | Fok1-nuclease | Fok1-nuclease | Cas9 nuclease |
| Target site | LAGLIDADG proteins | Zinc-finger binding sites | RVD tandem repeat region of TALE protein | PAM/spacer sequence |
| Recognition sequence size | 12–45 bp | 9–18 bp | 14–20 bp | 3–8 bp/20 bp |
| Targeting limitations | MN cleaving site | Difficult to target non-G-rich sites | 5′ targeted base must be a T for each TALEN monomer | The targeted site must precede a PAM sequence |
| Advantage | High specificity; Relatively easy to deliver in vivo | Small protein size; Relatively easy in vivo delivery | High specificity; Relatively easy to engineer; targets mitochondrial DNA more efficiently and causes fewer off-target effects than MNs and ZFNs. | Easy to engineer; Easy to multiplex |
| Disadvantage | Target locus must be put into the genome; complex to construct; difficult to multiplex; ineffectiveness and potential genotoxicity. The targeted locus must also contain the unique cleavage site for each endonuclease. | Expensive, time-consuming, labor-intensive, difficult to choose the target sequence, needing the coding gene to be custom-built for each target site, and highly off-target gene editing. All ZF domains must also be active. | Challenging to multiplex; not relevant in the case of DNA methylcytosine; a few in vivo deliveries; We should all be engaged; TALEs are nevertheless constrained by their repetitive sequences, which make it difficult to construct them using polymerase chain reaction (PCR), and by the fact that they are unable to target methylated DNA due to the possibility that cytosine methylation will impede TALE binding and alter recognition by its typical RVD. | Lower specificity; Limited in vivo delivery |

**Table 3.** *Cont.*

| Attribute | Meganucleases | Zinc Finger Nucleases | TALENs | CRISPR/Cas9 |
|---|---|---|---|---|
| DNA-recognition mechanism | HR-introduced Protein-DNA interactions | DSB-introduced by Protein-DNA interactions | DSB-introduced by Protein-DNA interactions | DSB introduced by RNA-guided protein-DNA interactions |
| Target specificity | High Positional mismatches are only occasionally accepted. Protein engineering is necessary for re-targeting. | High preference for G-rich sequences Positional mismatches are only occasionally accepted. Protein engineering is necessary for re-targeting. | High Requires a T at each of its target's five ends. Some positional inconsistencies are accepted. Retargeting necessitates intricate molecular cloning. | Moderate The two base pairs that PAM recognizes must come before the RNA-targeted sequence. Positional mismatches are only occasionally accepted. A new RNA guide is necessary for re-targeting. There is no need for protein engineering. |
| Multiplexing | + | + | + | ++++ |
| Delivery | accessible by transduction of viral vectors and electroporation | accessible by viral vector transduction and electroporation | simple in vitro conception Due to TALEN DNA's size and the recombination likelihood, it is challenging in vivo. | simple in vitro The big Cas9's inadequate packing by viral vectors is the cause of the mild difficulties of distribution in vivo. |
| Use as a gene activator | No | Yes endogenous gene activation minimal impacts off-target To target specific sequences, engineering work may be necessary | Yes endogenous gene activation has minimal impacts on off-target There are no time restrictions. | Yes endogenous gene activation minimal impacts off-target "NGG" PAM is necessary adjacent to the target sequence. |
| Use as gene inhibitor | No | Yes Works by repressing chromatin to prevent transcriptional elongation. minimal impacts off-target To target specific sequences, engineering work may be necessary. | Yes Works by repressing chromatin to prevent transcriptional elongation. minimal impacts off-target There are no time restrictions. | Yes Works by repressing chromatin to prevent transcriptional elongation. minimal impacts off-target "NGG" PAM is necessary adjacent to the target sequence. |
| Cost | High | High | High | Reasonable |
| Popularity | Low | Low | Moderate | High |
| Online resources | Database and Engineering for LAGLIDADG Homing Endonucleases (http://homingendonuclease.net/, accessed on 7 July 2021) | The Zinc Finger consortiums include software tools and protocols (http://www.zincfingers.org/, accessed on 7 July 2021) ZFNGenome—resources for locating ZFN target sites (https://bindr.gdcb.iastate.edu/ZFNGenome/, accessed on 7 July 2021) | Mojo Hand (http://www.talendesign.org/, accessed on 7 July 2021) or E-TALEN (http://www.e-talen.org/E-TALEN/, accessed on 7 July 2021) for TALEN design CHOPCHOP (https://chopchop.cbu.uib.no/, accessed on 7 July 2021) target site selection | Guide design: Zlab (https://zlab.bio/guide-design-resources, accessed on CRISPOR: http://crispor.tefor.net/, accessed on 7 July 2021; Benchling: https://www.benchling.com, accessed on 7 July 2021 AddGene: https://www.addgene.org/crispr, accessed on 7 July 2021; https://crispr.bme.gatech.edu, accessed on 7 July 2021 http://www.rgenome.net/cas-offinder/, accessed on 7 July 2021 |

### 4.2. Prime Editing

There is ongoing research into next-generation gene editing tools that promise increased precision and versatility. Base editors, for instance, enable the direct conversion of one base pair to another without inducing double-strand breaks, potentially minimizing off-target effects [133]. Prime editing, another recently developed technique, allows for the insertion, deletion, and substitution of base pairs with unprecedented precision [134].

Prime editing [135] entails small insertions, deletions, and base editing of single bases. Prime editing promises greater specificity by reducing off-target effects by requiring two additional nucleic acid matching steps because it uses CRISPR/Cas in addition to reverse transcriptase. The size of the base pairs that can be edited is constrained though, and more information about reverse transcriptase's error rate is still needed. Utilizing Cpf-1, a microbial nuclease preferred over Cas9 requires only one CRISPR guide RNA for specificity while producing staggering double-stranded DNA cuts instead of blunt ones is a change.

*4.3. Base Editing (BE)*

Current cytosine or adenine base editors can only accomplish C-to-T (G-to-A) or A-to-G (T-to-C) substitutions in the windows of target genomic sites of organisms; therefore, there is a need to develop base editors that can simultaneously achieve C-to-T and A-to-G substitutions at the targeting site [136]. However, they cannot correct variations other than these six transition mutations or other changes made by Prime editing, like DNA fragment insertions and deletions (PE). PE uses the prime editing guide RNA, a modified sgRNA, and a modified Cas9 nickase fused to reverse transcriptase (RT) enzyme (pegRNA). Clinical trials for base prime editing have yet to begin. However, conventional CRISPR/Cas9 and base editing share some characteristics with PAM restriction, off-targets, and delivery-impairing large molecules present in prime editing.

CRISPR applications have low homology-directed repair (HDR) editing efficiency compared to nonhomologous end joining. The Cytosine Base Editor (CBE), Adenosine Base Editor (ABE), and Glycosylase Base Editors (GBE) theoretically can function in both dividing and non-dividing cells to address several single-nucleotide polymorphisms (SNPs) linked to human diseases this inefficiency is, however, overcome by these base editors.

Base editing is a recent advancement in genome engineering. It represents a transformative approach that allows for precisely converting one DNA base pair to another without requiring double-strand breaks (DSBs) or donor DNA templates. The approach essentially involves using "base editors", which are proteins that can chemically modify the structure of a base, converting it into another base. This has numerous advantages over traditional genome editing techniques like CRISPR-Cas9.

BEs are mainly used to target point mutations that may result in an altered DNA sequence with novel or enhanced functions and gene inactivation requiring two main components: a Cas9n fused with a deaminase and an sgRNA that binds to a specific DNA sequence. Only one trial is listed on Clinicaltrials.gov, accessed on 10 June 2023 [137], with three totals in other databases [138].

The BE system primarily influences the editing window of nearby sites, off-target mutation, and product purity. The editing window should be minimized to increase target base accuracy when only one specific base pair needs to be changed accurately. On the other hand, a large editing activity window is used when the cytosine-based editor (CBE) system is used to introduce premature stop codons, produce large-scale saturation mutations, screen gene function, locate key amino acid positions in protein domains, etc.

One of the key advantages of base editing over traditional CRISPR/Cas9 methods is its precision. Instead of introducing DSBs, which can lead to various unintended changes in the genome, base editors can convert one base to another directly, offering more predictable and precise outcomes [139]. Many genetic diseases arise from point mutations—single base pair changes in the DNA sequence. Base editing offers the potential to correct these mutations directly [133]. Due to the absence of DSBs in base editing, there is a reduced risk of off-target mutations and chromosomal rearrangements, making the process safer for therapeutic applications [140].

Beyond therapeutics, base editing has considerable implications in research, from creating specific animal models with point mutations to studying the role of DNA modifications in epigenetic processes [141].

Base editors have a specific "editing window" or a range within which they can effectively modify bases. This restricts their applicability to target sequences that fall within this window. Some base editors can produce unwanted byproducts or edits, such as cytosine deamination outside the target site [142].

In conclusion, while base editing offers an innovative way to make precise changes to the genome, it is essential to be aware of its limitations. As technology is refined and its nuances understood, it promises to be a significant tool in therapeutic applications and biological research.

*4.4. Mitochondrial Base Editing*

Mitochondrial genome engineering is a rapidly advancing field due to the realization that mitochondrial DNA (mtDNA) mutations play a role in various diseases, from primary mitochondrial disorders to complex diseases like neurodegenerative conditions, cardiomyopathies, and even aging.

CRISPR-associated proteins transiently unwind dsDNA, enabling single-stranded DNA (ssDNA)-specific effector domains, such as APOBEC1 [143], to act locally on DNA sequences complementary to the gRNA's bound bases and generate CRISPR base editors [133]. This key feature of CRISPR-associated proteins has been widely exploited for the precise editing of nDNA, utilizing ssDNA-specific deaminases as accessory effector domains. Since a reproducible method for efficiently importing gRNAs into the mitochondrial matrix has remained elusive, CRISPR-based technologies cannot be effectively utilized for mtDNA editing. However, other programmable DNA-binding proteins, such as TALEs, can be efficiently imported into mitochondria, although they do not intrinsically unwind dsDNA. Therefore, mtDNA base editing was first accomplished using a novel protein capable of acting as a dsDNA-specific deaminase in conjunction with TALEs [144].

The discovery and characterization of double-stranded DNA deaminase A (DddA), a dsDNA-specific cytidine deaminase from *Burkholderia cenocepacia*, revolutionized the field of mtDNA editing [145].

Canonical DdCBEs consist of two arms, each comprising a TALE protein fused to the N- or C-terminus of $DddA_{tox}$ (the deaminase domain of DddA), followed by a uracil glycosylase inhibitor (UGI). Following well-established rules for the design of TALEs, these can be custom-made to target specific sequences within mtDNA. The ensuing TALE-mtDNA interactions bring both arms of a DdCBE pair into proximity, enabling the targeted reassembly of active $DddA_{tox}$. Subsequently, cytosine residues in a 5′-TC context and within the spacer region, i.e., the sequence between the two TALE binding sites, are converted to uracil. Provided that UGI impedes the excision of the resulting uracil residues, U•G intermediates are resolved into T•A base pairs during mtDNA replication, which occurs even in post-mitotic cells. This process results in programmed C•G-to-T•A conversions in the mitochondrial genome.

Repairing or altering the mitochondrial genome in vivo offers a compelling opportunity for treating these conditions. Mitochondrial diseases caused by mutations in mtDNA could be directly treated by repairing the affected sequences. Mitochondrial dysfunction is associated with various age-related diseases. Maintaining mitochondrial health through genetic interventions might delay the onset of these diseases. Modifying the mitochondrial genome might lead to enhanced cellular ATP production, benefiting conditions related to energy deficits.

TALENS and CRISPR/Cas9 genome-editing technologies have been adapted to target mtDNA. There have been some successes, especially with TALENs, in targeting and modifying mtDNA in cell lines [146,147].

MitoTALENs involve modifying TALENs to specifically target the mitochondrial genome, allowing the elimination of mutated mtDNA, leaving the healthy mtDNA to replicate and replenish the cell [148]. Loop Targeting involves the D-loop, a region in the mitochondrial genome with high transcriptional activity. Targeting this area can lead to the degradation of the entire mitochondrial genome, allowing for potential "mtDNA transplantation" [149].

As with all genome editing, there is a risk of off-target mutations, which might be detrimental. Most cells have a mixture of healthy and mutated mtDNA, known as heteroplasmy. Managing and ensuring that edited mtDNA outcompetes and replaces the diseased mtDNA in every cell is challenging. Efficiently delivering the genome editing tools to mitochondria in every cell of an organism is a significant challenge.

## 4.5. Nickase-Based Genome Engineering

Inadequate outcomes, such as p53 activation, translocations, off-target mutations, and complex undesired products, could be linked to DSBs at targeted genomic loci. Since point mutations account for half of all known disease-associated gene variants, the Cas9 nickases have become essential tools with a targetable property. One of the two Cas9 nuclease domains is modified to produce the CRISPR nickase. Nickases can reduce the likelihood of off-target editing by causing a single-strand break instead of a double-strand break when used with two adjacent gRNAs. The Cas9 variants Cas9n D10A and Cas9n H840A mediate the cleavage of a single DNA strand in the complementary or non-complementary DNA strand of gRNA.

## 4.6. Homologous Recombination

Homologous recombination is a fundamental cellular process that ensures genomic integrity and diversity during cell division, particularly during meiosis and mitosis. In this process, DNA sequences are exchanged between two similar or identical strands of DNA. Below, I explain how homologous recombination occurs and offer some examples to elucidate the concept further.

Strand Exchange Proteins: The RecA protein in bacteria and Rad51 in eukaryotes are essential for strand invasion and forming the initial joint molecule. Rad51 polymerizes along the single-stranded DNA (ssDNA), and this nucleoprotein filament searches for a homologous sequence on the sister chromatid or homologous chromosome [150].

Holliday Junction Dynamics: The Holliday junction can migrate along the DNA, allowing an extended sequence of genetic material to be exchanged. Resolution of the Holliday junction can occur in vertical or horizontal orientations. The orientation determines whether crossover (exchange of flanking markers) or non-crossover (no exchange of flanking markers) products are formed [151].

Regulation: In eukaryotes, homologous recombination is highly regulated to ensure it occurs only at the right time and place. For example, the tumor suppressor gene BRCA2 plays a vital role in recruiting Rad51 to the site of damage [152].

Meiosis and Genetic Diversity: In humans and other sexually reproducing organisms, homologous recombination during meiosis creates new combinations of alleles, which are passed on to offspring. This process is the basis for the genetic variability necessary for natural selection and evolution. The Spo11 protein initiates the double-strand breaks, and a cascade of proteins like Rad51 and Dmc1 facilitates strand invasion and Holliday junction formation [153].

DNA Repair and Cancer: Defects in homologous recombination can lead to genome instability and, ultimately, cancer. WHEN MUTATED, the BRCA1 and BRCA2 genes significantly increase the risk of breast and ovarian cancers, partly because of their role in homologous recombination-based DNA repair [154].

Bacterial Evolution: Bacteria can acquire antibiotic-resistance genes from other bacterial strains through homologous recombination. This has implications for the development and spread of antibiotic resistance [155].

The exchange of nucleotide sequences between two DNA molecules that are similar or identical is referred to as homologous recombination in genetics. This is frequently used in mouse genetics. This technique, also known as gene targeting, allows for precisely replacing a gene copy by integrating a gene distinct from the original gene. The development of knockout mice using embryonic stem cells to deliver synthetic genetic material to suppress the target mouse gene is of therapeutic interest. Additionally, this technology enables a reliable and effective knocking of a specific mutation, reporter, or human gene sequence into endogenous loci, producing precise and physiologically more accurate models of human disease [156].

*4.7. TFD-ODN Techniques*

One option to obstruct a known activated regulatory pathway that promotes disease is to target transcription factors. Therapeutic drug candidates called double-stranded transcription factor decoy (TFD) oligodeoxynucleotides (ODN) specifically target and neutralize the main transcription factors involved in the pathogenesis of a particular disease. The consensus DNA binding site of a particular transcription factor in the promoter region of its target genes is mimicked by these brief double-stranded TFD molecules. This nucleic acid-based drug class can treat diseases brought on by the aberrant expression of such target genes, whose byproducts are involved in disease initiation and progression. Specific drug delivery techniques, such as tissue-specific transduction using adeno-associated viral (AAV) vectors or long-term TFD molecule expression in non-dividing cells using ultrasound-targeted microbubble destruction with TFD ODN-coated microbubbles, are also futuristic propositions.

*4.8. Argonautes*

Almost all eukaryotes, bacteria, and archaea contain members of the Argonautes gene family, which is highly conserved. The Argonaute gene family is found in many animal and plant genomes, but the nematode Caenorhabditis elegans stands out because it has at least 26 Argonaute genes. Four conserved domains are found in argonaute proteins: the N-terminal, PAZ (charged with short RNA binding), Mid, and PIWI (which confers catalytic activity). Argonaute proteins function in RNA-based silencing mechanisms by altering protein synthesis and influencing RNA stability. They associate with small non-coding RNAs, such as microRNAs and small interfering RNAs (siRNAs). Piwi-interacting (pi) RNAs, a novel class of small non-coding RNAs that help maintain chromosome integrity and are involved in the maturation of siRNA and miRNA, can also be produced by argonaute proteins [157].

*4.9. Integrase*

A retrovirus (like HIV) that forms covalent connections between its genetic material and the host cell it infects produces retroviral integrase (IN). These 288 amino acids 32 kDa viral enzyme mediates the linkage of double-stranded viral DNA into the host cell genome [158]. Retroviral INs should not be confused with phage integrases (recombinases) used in biotechnology, such as phage integrases discussed in site-specific recombination. The IN macromolecule is an essential part of the intasome. This macromolecular complex is bound to the ends of viral DNA and the retroviral pre-integration complex [159].

*4.10. Recombinase*

In multicellular organisms, DNA recombinases are frequently used to modify genome structure and control gene expression. These enzymes, known as bacteria phages, are derived from bacteria and fungi, and they catalyze DNA exchange reactions between short (30–40 nucleotide) target site sequences specific to each recombinase. Excision, insertion, inversion, translocation, and cassette exchange are the four fundamental functional modules made possible by these reactions. These modules have been used singly or in various combinations to control the expression of genes. Cre recombinase, Hin recombinase, Tre recombinase, and FLP recombinase are examples of recombinase types.

## 5. Current Status

With its ability to directly modify the genetic code, GE has the potential to revolutionize medicine, particularly in genetic diseases that account for most human diseases. The development of tools like CRISPR/Cas9 has made gene editing a practical approach for treating various human diseases caused by single-gene mutations [160,161].

A significant novelty in GE efforts comes from the safety and efficacy of CRISPR/Cas9 technology that has been at the forefront of these developments [123], where clinical trials have demonstrated that gene editing can increase the production of fetal hemoglobin in

the red blood cells of patients with β-thalassemia and sickle cell disease [44]. This approach has shown significant promise, with patients demonstrating clinical improvements and decreased need for transfusions [162]. Early successes include using AAV-delivered CRISPR/Cas9 components for treating Leber congenital amaurosis [163] and muscular dystrophy in animal models [53].

Similarly, promising preclinical results for treating Duchenne Muscular Dystrophy (DMD), a fatal muscle disorder, give much confidence in using the CRISPR/Cas9 approach to restore dystrophin expression in animal models of DMD, offering hope for future clinical applications [53].

While we have already seen early-stage clinical trials of gene editing therapies for certain diseases, such as sickle cell disease and β-thalassemia [44], many potential gene editing applications are still in the preclinical or early clinical development stage [164]. This underlines the need for more research to understand and improve the limitations of current technologies, such as improving the precision of gene editing, optimizing delivery systems, minimizing off-target effects, and resolving ethical and safety concerns [165,166]. Regarding off-target surveillance, genetic diversity is important when designing and evaluating gene editing products [167].

## 6. Challenges and Concerns

Despite the enormous potential of gene editing technologies, significant challenges remain, including efficiency, precision, delivery, and safety issues [168].

Applications for GE have sparked questions in science, safety, ethics, and legislation. GE in human embryos is not supported by NIH funding. The Dickey–Wicker Amendment [169] forbids the use of monies appropriated by Congress for research involving the creation of human embryos or the destruction of human embryos. The international community has called for a moratorium on heritable gene editing (HGE) since the birth of twins in China in 2018 [170] that were genetically modified using CRISPR/Cas9. Several expert groups have been formed to create global governance frameworks [171]. The established [172] best practices require multi-stakeholder decision-making, including learned medical societies stating that "eugenic development of offspring or the genetic manipulation of non-disease traits may never be justifiable" [173]. Regulators should consider the need to support anti-editing tools like anti-CRISPR, which can prevent or undo unwanted gene editing, to counteract the misuse of gene editing tools [174]. However, there is a growing concern about these limitations that might slow the progress of GE technologies and the abuse of use technologies [175].

A significant issue in GE pertains to biohacking, which presents serious health issues, left with little control despite many incidences of self-medication and even producing gene-edited babies. Additionally, the widely available DIY kits pose a severe risk of off-target editing, unintended off-target editing consequences, and unknowable long-term off-target editing effects [176].

Given these risks, the NIH guideline [177] to streamline gene therapy was established. However, in a recent revision of this guideline, the NIH dropped the requirement to register and report on human gene therapy protocols. As a result, only the FDA has regulatory oversight of all human GT and GE, treating them like any other biological product. The NIH now focuses on safety and ethical issues [178,179], yet much of the GE remains uncontrolled.

Gene editing has also shown promise in treating infectious diseases, particularly in the case of HIV, where this retrovirus integrates its genome into host cells, making it extremely difficult to eradicate from the body. However, with gene editing, researchers have been able to excise the integrated HIV genome from infected cells in vitro [180].

Despite much optimism, several challenges must be addressed before these therapies can be broadly and safely applied in the clinic [181].

Off-target Effects: One such challenge is the potential for off-target effects, i.e., unintended modifications at sites other than the target locus. Such off-target edits could

potentially lead to harmful consequences, including the activation of oncogenes or the inactivation of tumor suppressor genes, resulting in cancer. While CRISPR and other gene-editing technologies are precise, they are not perfect. Unintended genetic modifications can have unknown and potentially dangerous consequences [123,182]. Methods to improve the specificity of gene editing enzymes and to better detect and predict off-target effects are being actively researched. The development of high-fidelity versions of Cas9, such as eSpCas9 and SpCas9-HF1, represents significant strides in this direction [183].

Chromosome Loss: CRISPR-Cas9 genome editing has enabled advanced T-cell therapies, but occasional loss of the targeted chromosome remains a safety concern. Arrayed and pooled CRISPR screens reveal that chromosome loss is generalizable across the genome and results in partial and entire chromosome loss, including in pre-clinical chimeric antigen receptor T cells. The T cells with chromosome loss persist for weeks in culture, implying the potential to interfere with clinical use. In a first-in-human clinical trial of Cas9-engineered T cells, a modified cell manufacturing process has shown dramatically reduced chromosome loss while essentially preserving genome editing efficacy. Expression of p53 correlates with protection from chromosome loss observed, suggesting both a mechanism and strategy for T cell engineering that mitigates this genotoxicity in the clinic These recent findings need closer attention to the safety issues in gene editing [184].

Immune Response: Another challenge is the potential immune response to gene editing components, particularly the bacterial-derived Cas9 protein. Pre-existing immunity to Cas9 could reduce the efficacy of gene editing. It could also lead to serious adverse effects [185]. Approaches to mitigate this immune response, such as using immunosuppressants or developing humanized versions of Cas9, are being explored [186].

Delivery Systems: The efficient and safe delivery of gene editing components to target cells in the body is another critical challenge. Several strategies are being explored, including viral vectors, lipid nanoparticles, and cell-penetrating peptides [187]. However, each approach has limitations, such as immunogenicity, limited cargo capacity, and off-target tissue distribution [188]. Novel delivery strategies, such as engineered exosomes [189] or gold nanoparticles [190], are currently under investigation and could potentially overcome some limitations. The future of gene editing, potentially revolutionizing the treatment of a wide range of diseases, including many that currently have no cure [191], and many challenges remain. Technical challenges include improving gene editing tools' efficiency, precision, and safety. While significant progress has been made in developing more precise and efficient gene editing technologies, further improvements are needed to minimize off-target effects and ensure the safety of these therapies [4]. Developing delivery systems that selectively target specific tissues and cells is another important area of research [187].

Heterogeneity-related Testing Limitations: Unlike drug actions that are anticipated based on preclinical studies, including the phase 1 and 2 data, the GE tool actions are highly individualistic, and in several situations, larger-scale clinical trials may not conform to the conclusions drawn from earlier studies. However, this heterogeneity should be expected, though it is not the normal perspective in drug development. For example, an in vivo CRISPR/Cas9 genome editing to treat Leber congenital amaurosis 10 (LCA10) demonstrated a favorable safety profile across all dose cohorts; phase 3 studies did not correlate, and the study was halted [192]. Another example relates to a recent report that Zolgensma, launched in 2019 as a "potential cure" for spinal muscular atrophy (SMA), the most expensive drug then, as a single dose rather than continuous treatment in the world at the time, did not show homogenous responses [193].

Ethical Concerns: Ethical challenges relate to issues such as germline editing, consent, and the potential use of gene editing for enhancement purposes. These issues will require ongoing discussion and consensus-building among scientists, ethicists, policymakers, and the public [6].

Germline Editing: One of the primary ethical issues is the potential for "germline" editing, which involves modifications to the DNA of sperm, eggs, or embryos. These changes would affect the individuals they are made to and their descendants [31]. While

this could provide an avenue to eliminate genetic diseases from family lines, it also opens a range of ethical considerations, including the potential for unintended consequences and the implications of permanently altering the human gene pool [6]. In addition to germline editing, "somatic" gene editing—which involves changing the genes in cells of a specific tissue in an individual and does not affect future generations—also raises ethical issues. These primarily revolve around consent, particularly in scenarios where the technology is used in children or other individuals unable to provide informed consent [194]. Since germline editing changes are passed to future generations, this introduces debates about the potential for "designer babies" and unintended long-term consequences. Ethical considerations are another significant aspect of gene editing technologies, particularly concerning germline editing, i.e., modifications that can be passed on to subsequent generations [195]. The controversy surrounding the announcement of the birth of gene-edited babies in China in 2018 highlights the urgency of addressing these ethical concerns [196].

Augmentation: The possibility of gene editing being used for enhancement rather than therapeutic purposes—such as altering physical traits or abilities—also presents a significant ethical challenge. It raises questions about the nature of human beings and the acceptability of designing our species [197].

Regulation and Oversight: Ensuring gene editing technologies are used safely and ethically requires robust regulations and international cooperation. Ethicists, scientists, and policymakers worldwide are actively engaged in discussions to develop guidelines and regulations for the ethical use of gene editing technologies. Notably, the World Health Organization has established an expert advisory committee to develop global standards for governance and oversight of human genome editing [198].

Technical Limitations: While gene editing is powerful, delivering the editing tools into specific cell types or tissues, especially in adult organisms, remains a challenge.

Public Perception: Public understanding and acceptance of gene editing vary, and misinformation or misconceptions can influence regulatory decisions [199].

Economic and Access Issues: Even if therapies are developed using gene editing, ensuring they are affordable and accessible to all who need them is a significant concern. Advanced therapies may be developed, but it is crucial to ensure they do not widen the gap between the rich and the poor. One issue pertains to the experimental use of NGS. With the increasing integration of NGS into clinical practice, reimbursement becomes crucial. Since NGS can lead to better, more tailored treatments and prevent costly interventions for incorrect diagnoses, insurance companies, and health systems worldwide are starting to acknowledge its value. However, reimbursement policies vary by country and even within regions of countries. In the U.S., the Centers for Medicare and Medicaid Services (CMS) has proposed coverage for NGS for patients with advanced cancer. This marks a significant step toward broader coverage for genetic tests [200]. Several studies have evaluated the cost-effectiveness of NGS in various diagnostic scenarios, emphasizing that while the upfront costs might be higher, the long-term savings from precise, targeted treatments and interventions can be substantial [201]. Despite the evident benefits, challenges like the interpretation of variants of unknown significance, data storage, and the necessity of genetic counseling can influence reimbursement policies [202]. In conclusion, as NGS continues to play an increasingly significant role in diagnostics and personalized medicine, its reimbursement by healthcare systems will likely evolve. Continued studies on its cost-effectiveness and value in improving patient outcomes will be pivotal in shaping these policies.

Long-term Effects: Especially when considering germline editing, the long-term effects on individuals and the potential impact on the human gene pool are unknown.

Biosecurity Concerns: There are concerns that gene editing could be misused, intentionally or unintentionally, leading to bioterrorism threats or ecological imbalances [203].

Cultural and Social Impacts: The broad use of gene editing might have cultural consequences, changing how societies view disabilities, diseases, or even certain human traits.

Informed Consent: Particularly in the context of germline editing, future generations who have been edited cannot give consent, raising ethical concerns.

Unforeseen Ecological Consequences: When modifying organisms that are then released into the wild, there is a risk of unintended ecological consequences [204].

Unpredictable Outcomes: While gene editing tools are precise, sometimes off-target effects or unintended consequences within the genome can occur.

Genetic Homogenization: If gene editing is used broadly, especially in a reproductive context, it might decrease genetic diversity, which is crucial for the robustness and adaptability of populations.

Regulation and Oversight: Striking a balance between innovation and safety is challenging. Too little regulation might expose individuals to risks, while too much might stifle potentially life-saving innovations [123]. The rapid advancement of gene editing technologies necessitates equally rapid development of regulatory guidelines to ensure safety and efficacy. Regulatory bodies worldwide are currently grappling with the challenge of developing robust frameworks for gene editing therapies [205,206].

Affordability: The issue of equity is a crucial ethical concern, given the potential for gene editing to be accessible only to the wealthy, thus exacerbating existing health and social inequalities [207].

Global Disparity: There are significant disparities in access to advanced healthcare technologies across different regions, which could be exacerbated with the advent of gene editing therapies [208]. For instance, patients in low- and middle-income countries may face significant barriers to accessing these therapies due to a lack of healthcare infrastructure, financial constraints, and regulatory challenges [209]. Addressing these challenges would require concerted efforts from governments, policymakers, healthcare providers, and industry stakeholders. For instance, policies could be developed to incentivize the production of affordable gene editing therapies and facilitate access to needy patients [210].

Healthcare Infrastructure: Moreover, robust healthcare infrastructure would need to be developed in regions with limited access to advanced healthcare technologies. This would entail investments in healthcare facilities, training healthcare professionals, and evolving regulatory frameworks to ensure gene editing therapies' safe and effective use [211]. About 400 million people worldwide with 7000 diseases caused by mutations in single genes [212] can benefit if GE products are made affordable. Gene editing needs are continuously expanding, such as the recent suggestions to monitor children for prospective genetic disorders and fix them well before they become evident. In terms of the market, the global gene editing market was valued at approximately $3.7 billion in 2019, and it is projected to reach $8.1 billion by 2025. The major players in the market include companies like CRISPR Therapeutics, Intellia Therapeutics, Editas Medicine, and Sangamo Therapeutics [213].

## 7. Regulatory

The legal and regulatory landscape surrounding gene editing is complex and rapidly evolving. Current regulatory frameworks worldwide differ significantly in their approach to gene editing, with some countries allowing for broad use in research and others restricting its use, particularly for germline editing [214].

The Food and Drug Administration (FDA) regulates gene editing in the United States under its broader oversight of gene therapy. The FDA has laid down guidelines for preclinical and clinical development of gene therapy products, which include gene editing therapies. The agency requires rigorous testing for safety and efficacy before such treatments can be approved for use in patients [215].

The European Medicines Agency (EMA) regulates gene therapies, including gene editing, in the European Union. The EMA has a similar approach to the FDA, requiring comprehensive preclinical and clinical testing for safety and efficacy. Additionally, the EMA considers the potential environmental impact of gene editing, particularly for germline modifications [216].

In many Asian countries, such as China and Japan, gene editing regulations are less stringent. This has led to rapid advancements in gene editing research in these countries and raised ethical and safety concerns [217].

Overall, the regulatory framework for gene editing must balance the need for innovation with ethical, safety, and societal considerations. Developing international standards and guidelines could help harmonize the global approach to gene editing and ensure that it is used responsibly and for all benefits [218].

Regulatory challenges involve creating a regulatory framework that can keep pace with the rapid advancements in gene editing technology while ensuring the safety and efficacy of gene editing therapies. Harmonizing regulatory standards across different countries is another critical challenge to address [214].

Regulators can do more to promote the standardization of off-target (and on-target) effect measurement, including implementing the proper methods, sample handling practices, quality control measures, data analysis, and clinical interpretation. For example, the EU guidelines for genetically modified cells' quality, non-clinical requirements, and clinical requirements do not specify methods for determining on- and off-target effects because they are still evolving [219]. The FDA's guidance is similar in not outlining how off-target effects discovered in pre-clinical studies should be monitored over an extended period [220].

Since gene editing is a newer field, the regulatory control standards take a high risk-based approach, in the abundance of caution that has significantly hampered the entry of GE products. In certain situations, relying solely on product quality results will not be appropriate, requiring additional non-clinical and clinical data. In addition, release testing may be affected by the personalized nature of gene editing therapies, regulatory batch testing, and release requirements, which can consume a sizable portion of the batch.

Due to the significance of the in vivo cellular environment on gene editing efficiency, for instance, there are difficulties in predicting the clinical effects of gene editing treatments in humans using non-clinical efficacy models. A thorough investigation and attempts to develop such relevant non-clinical efficacy models should be discussed in any case and requested on a case-by-case basis in regulatory interactions, despite the idea that gene editing treatments may not need them or may not have any relevant non-clinical efficacy models.

Numerous unknowns exist regarding the handling and regulatory classification of gene editing products. Some gene editing products, for instance, are subject to genetically modified organism (GMO) regulation and requirements in the EU [221], where various competent authorities assess an ATMP GMO submission and a Clinical Trial Application (CTA) submission, resulting in timing and content inconsistencies. Several initiatives, such as standard application forms and good practice documents, are being made to clarify and harmonize the requirements [222].

A Guideline on Assuring the Quality and Safety of Gene Therapy Products has been published by the Pharmaceuticals and Medical Devices Agency (PMDA, Japan) (not specific to gene editing). However, this raises scientific and policy concerns. Therefore, to maintain the orphan status of potentially curative ATMPs, significant benefit over an authorized ATMP must be demonstrated. Most ex vivo GE products will fulfill the ATMP definition as Gene Therapy Medicinal Products (GTMPs, European Medicines Agency) or cell therapies. The US situation is more precise because the FDA classifies all gene editing products (both in vivo and ex vivo) as gene therapy products.

Gene editing products may also have difficulties maintaining orphan designation and proving significant benefits because of their long-lasting or potentially curative effects. In addition, clinical efficacy data to demonstrate superiority may not be attainable for scientific reasons, leaving clinical safety or even only non-clinical data as the primary evidence for a significant benefit claim.

## 8. Delivery Tools

The development of in vivo gene editing delivery methods remains an area of rapid progress, with ongoing research aimed at overcoming the existing challenges for each system, such as specificity, efficiency, and safety.

### 8.1. Overview

Gene editing has been revolutionized by technologies like CRISPR-Cas9, TALENs, and ZFNs, yet the successful application of these tools in vivo remains contingent upon effective delivery systems. Several innovative strategies have been developed to address this challenge, each with its advantages and drawbacks.

Recombinant adeno-associated virus (AAV) vectors have been, or are currently in use, in 332 phase I/II/III clinical trials in several human diseases. In some cases, remarkable clinical efficacy has also been achieved. There are now only three US Food and Drug Administration (FDA)-approved AAV "drugs", bringing the awareness that the first generation of AAV vectors is not optimal. One drawback is when relatively large vector doses are needed to achieve clinical efficacy, resulting in host immune responses culminating in serious adverse events and, more recently, in the deaths of 10 patients [223].

Viruses like AAV (Adeno-Associated Virus) remain among the most widely used delivery systems. AAVs can efficiently target various tissues, and numerous serotypes allow for tissue-specific targeting [224]. However, the risk of immunogenicity is an ever-present concern [225]. Lentiviruses have also been employed due to their ability to integrate into the host genome and thus offer long-term expression of the delivered gene. Yet, random integration can lead to insertional mutagenesis, posing safety concerns [226].

Non-viral vectors offer alternative solutions to these challenges. Lipid nanoparticles (LNPs) encapsulate the gene-editing machinery and protect it from enzymatic degradation [227]. Moreover, LNPs can be functionalized with ligands that facilitate targeted delivery to specific cell types [228]. However, toxicity and off-target effects remain issues [229]. Similarly, gold nanoparticles have shown promise due to their biocompatibility and capacity for surface modification [190]. Their major drawback is the possible accumulation in organs like the liver and spleen, which can lead to long-term toxicity [230].

Physical methods like electroporation temporarily destabilize the cell membrane using an electrical pulse, facilitating the entry of the gene-editing tools [231]. While highly efficient, the procedure can be stressful to the cells and may result in cell death [232]. Hydrodynamic injection, a technique involving rapid, large-volume intravenous injections, offers an alternative but is generally limited to liver-targeted delivery [233].

Increasingly sophisticated methods are also being explored. Exosome-based delivery utilizes naturally occurring vesicles to package and deliver gene-editing tools [234]. This method offers the advantage of evading the immune system, although the scalability for clinical use is still under investigation [235]. Immune-cell conjugation directly binds gene-editing tools to immune cells for targeted delivery, holding promise, especially in the context of immunotherapies [236].

Advanced technologies like light-induced systems and magnetofection offer unprecedented control over the spatiotemporal aspects of delivery. Light-induced systems use light-sensitive molecules that release the gene-editing tools upon exposure to specific light wavelengths [237]. Magnetofection employs magnetic nanoparticles guided by external magnetic fields to deliver the gene-editing machinery to the desired location [238].

Each of these delivery systems has its challenges, such as immunogenicity, off-target effects, or technical complexity. Furthermore, many of these methods are still experimental and may face regulatory hurdles before they can be routinely applied in clinical settings. However, ongoing research continually provides novel solutions that improve the safety, efficiency, and specificity of in vivo gene-editing delivery systems [239].

Magnetic Resonance Targeting (MRT) offers a non-invasive technique to guide gene-editing delivery. This approach uses magnetic fields to direct magnetic nanoparticles carrying

the gene-editing tools to specific locations within the body [240]. Despite its potential, MRT is currently constrained by the need for specialized equipment and expertise [241].

Scaffold-mediated delivery is another emerging approach where the gene-editing elements are immobilized onto biodegradable scaffolds implanted into the target tissue. The controlled scaffold degradation over time allows for the gradual release of the gene-editing machinery [242]. While promising for localized applications like bone and cartilage repair, the invasive nature of the method can be a limitation [243].

Chemically inducible systems use small molecules to control the activity of gene-editing enzymes. These systems can be engineered to only become active in the presence of a specific molecule, which can be administered orally or through injection, allowing for temporal control over gene editing [244]. While promising precise control over gene editing, the method requires the concurrent development of specific inducer molecules [245].

The aptamer-based delivery systems offer a highly specific yet adaptable approach for delivering gene-editing machinery. Aptamers are short, single-stranded DNA or RNA molecules that can bind to various targets with high specificity. Attaching aptamers to gene-editing tools makes targeting specific cell types or even intracellular compartments possible [246]. However, the challenge lies in identifying aptamers with high specificity and affinity for the target [247].

Each of these methods is in varying stages of research and development, with many facing hurdles such as scalability, immunogenicity, or off-target effects that must be overcome before clinical application. Nevertheless, the ongoing advancements in in vivo delivery systems are promising for the future of gene editing therapies, making it an active and rapidly evolving field of research [248].

Peptide-Mediated Delivery involves the use of specific peptides to facilitate the delivery of gene-editing components into cells. Cell-penetrating peptides (CPPs) can be conjugated to the gene-editing machinery to improve uptake [249]. This technique shows promise but may be limited by the cargo size and endosomal entrapment issues [250].

Sonoporation uses ultrasonic waves to create temporary pores in cell membranes, allowing for the entry of gene-editing components [251]. It is an attractive option due to its non-invasive nature; however, there are challenges around controlling targeting specificity [252].

Light-inducible systems leverage optogenetic principles to activate or deliver gene-editing elements in response to specific wavelengths of light. Such systems offer exact spatiotemporal control [253], but their application in deep tissues is limited due to the poor penetration of light [254].

Hydrogel-based Delivery employs hydrogels that can release gene-editing components in a controlled manner after being injected into specific tissues. This has advantages in localized delivery but raises concerns about the complete biocompatibility of the hydrogels used [255].

Viral-like particles (VLPs) are non-infectious nanoparticles resembling viral capsids that can encapsulate gene-editing tools. While they can offer high efficiency and specificity, concerns remain about their immunogenicity and potential to trigger unintended biological responses [256].

Microfluidic Devices offer another potential delivery avenue. These devices can isolate and manipulate individual cells, facilitating the direct injection of gene-editing components. Though technologically challenging and generally suited for ex vivo applications, advances are being made to adapt this technology for in vivo use [257].

Electrostatic complexes form between gene-editing components and polycations, which facilitate cellular uptake. These complexes are relatively easy to prepare but often lack the specificity required for targeted delivery [258].

Each delivery system has advantages and limitations, making the choice highly dependent on the specific application and therapeutic context. The field of in vivo gene-editing delivery systems is incredibly dynamic, with new techniques continually emerging [259].

Exosome-mediated delivery: Exosomes are small vesicles secreted by cells that can encapsulate and transfer bioactive molecules, including gene-editing components. This method offers the advantage of natural targeting abilities and reduced immunogenicity [234]. However, loading efficiency and scalability remain challenges [260].

Ultrasound-targeted microbubble destruction (UTMD): This technique involves encapsulating gene-editing tools in lipid microbubbles that are destroyed using focused ultrasound to release their payload. This allows for spatially targeted delivery but may induce tissue damage at the release site [261].

Lipid polymer hybrid nanoparticles: These are complex nanostructures that combine the merits of polymeric and lipid-based delivery systems. These nanoparticles can encapsulate gene-editing tools and offer stability, biocompatibility, and tunable release kinetics [262]. However, they are still in the early stages of development [263].

Mechanical pinpoint injection: This technique involves using fine needles to inject the gene-editing components directly into the target tissues mechanically. While invasive, this technique can achieve high levels of specificity and is currently used in some ocular gene therapy trials [264].

Gold nanoparticle-assisted delivery is functionalized to carry gene-editing molecules and facilitate their delivery into cells. This technique is promising due to its low cytotoxicity, but concerns exist regarding the potential long-term effects of gold accumulation [265].

RNA nanoparticles are constructed from RNA molecules and can carry gene-editing machinery for targeted delivery. RNA nanoparticles offer the advantage of easy modification for targeted delivery but have so far been limited by stability issues [266].

Prodrug systems comprise inactive or less active forms of the gene-editing tools that are delivered and activated in situ by specific enzymes or conditions. This allows for targeted delivery with reduced off-target effects [267].

Nanodiamond vectors are diamond nanoparticles functionalized to carry gene-editing molecules. They offer high payload capacity and lower toxicity but are still experimental [268].

While systemic delivery using targeting ligands is a suitable choice, specific delivery methods are disease-dependent; this includes local administration that can be very effective for treating conditions affecting muscle, skin, eye, and ear tissues. Other delivery methods include electroporation, plasmids, and nanovesicles, such as exosomes with different delivery profiles [269].

### 8.2. CRISPR

Regarding delivery, CRISPR-Cas can be divided into plasmids, large-sized biomacromolecules, mRNA/gRNA, and RNP. Numerous nonviral delivery methods successfully deliver CRISPR-Cas into ex vivo and in vivo target cells. These delivery systems primarily contain cationic lipids, cationic polymers, cationic polypeptides, DNA nanostructures, AuNPs, cell-derived vesicles, and other organic materials, as was previously mentioned.

Directly introducing sgRNA and Cas9 nuclease can rapidly knock out genes using electroporation, lipofectamine therapy, or injection.

### 8.3. Vectors

Many vectors used in gene therapy, which typically emphasizes long-term expression to correct genetic flaws, are rarely appropriate for GE, which only needs temporary delivery of editors. The most popular editors also present additional challenges because of their large sizes (SpyCas9 and TALENs), repetitive sequences, and need to deliver both components of a ribonucleoprotein complex (ZFNs and TALENs), RNP; for example, in CRISPR). Accurate tissue targeting is also necessary due to the possibility of on-target or off-target activity in the incorrect tissues [270].

### 8.4. AAV

The AAV vectors demonstrate tissue-specific tropism, immunogenicity, and tumorigenic risks that limit their clinical applications [271]. They produce neutralizing antibodies;

it works for vaccines but not for GE products. Although viral vectors are still the main delivery systems of in vivo genome editing, immunogenicity results in less genetic material reaching its destination and prohibits patients from receiving a second (or third) dose. Other vectors include adenoviruses, their associated viruses, and integrase-defective lentiviral vectors (IDLVs).

Since their discovery about 50 years ago, the replication-defective, non-pathogenic, almost universal single-stranded adeno-associated viruses (AAVs) have gained significance. AAVs are small viruses that are not currently known to cause disease. Because of their ability to deliver DNA to cells, they are often used as vectors to introduce or modify genes within an organism. AAVs are especially appealing because they target various cell types and tissues without inducing an immune response. When combined with tools like CRISPR/Cas9, AAVs can deliver the necessary components for gene editing into specific cells. This allows for targeted manipulation, which is especially useful in therapeutic contexts. For example, Luxturna, an FDA-approved gene therapy for a rare form of inherited blindness, uses AAV to deliver a functional copy of the RPE65 gene to retinal cells. Research is ongoing into using AAV vectors to deliver microdystrophins, shortened versions of the dystrophin gene, to muscle cells in DMD patients [272]. Researchers aim to correct genetic mutations responsible for certain liver diseases by delivering CRISPR components via AAV vectors to liver cells [273].

Recombinant AAVs were created as suitable genetic medicine tools and have now developed into efficient, marketable gene therapies thanks to their distinct life cycle and virus-cell interactions. The capacity of AAVs to accurately modify the genome sets them apart from other types of viruses. Furthermore, AAV only uses the high-fidelity homologous recombination (HR) route and does not require exogenous nucleases for the previous cleavage of genomic DNA, in contrast to all current GE platforms. This leads to an exact editing outcome that preserves genomic integrity without incorporating indel mutations or viral sequences at the target site while preventing off-target genotoxicity.

AAVs have a limited DNA-carrying capacity. This can be challenging when delivering larger genes or multiple components of gene-editing systems. Like any gene-editing approach, there is the potential for unintended changes to the genome, which can have unforeseen consequences. Despite AAV's generally low immunogenicity, there is potential for the host to develop an immune response, especially with repeated administration.

Stem cells, especially pluripotent stem cells (iPSCs or ESCs), can potentially become any cell type in the body. Using AAV vectors to introduce gene-editing tools like CRISPR/Cas9 into stem cells offers a two-fold opportunity; first, stem cells can be edited ex vivo (outside the body) to correct genetic mutations and then be transplanted back into the patient. Second, edited stem cells can differentiate into various tissue types, providing a sustained therapeutic effect in the body [273].

HSCs give rise to all blood cell types. AAV vectors have been explored to correct mutations in HSCs for disorders like sickle cell anemia. AAV-mediated delivery of gene-editing tools into iPSCs derived from patients with neurodegenerative diseases like Huntington's offers a potential method of creating disease-free neural cells for transplantation [274]. By using AAV-mediated gene editing in stem cells, researchers can potentially create "universal" stem cells lacking immunogenic markers, making them suitable for transplanting any patient without rejection [275]. While AAVs are efficient vectors, achieving high editing efficiency in stem cells remains challenging. There is the potential for unintended changes to the genome of stem cells, which could have harmful consequences. Ensuring edited stem cells differentiate into the desired cell types without forming tumors (teratomas) is crucial. In conclusion, the combination of AAV-mediated gene editing and stem cell therapeutics is a dynamic field with great promise. However, it still requires thorough research to ensure safety and efficiency [276].

The use of Herpes Simplex Virus (HSV) vectors for gene editing has generated interest in both academia and industry due to their unique properties. However, using HSV vectors in gene editing has both advantages and disadvantages. The pros include its high

payload capacity, allowing the insertion of large and complex gene constructs [277]; the load can be over ten times more genetic material than an AAV and over twice that of a lipid nanoparticle (LNP); since HSV vectors are fully neutralized in patients with prior exposure and almost 90% of the population is exposed to HSV in the US and EU, making multiple dosing of HSV is ethical and practical. This allows various doses of HSV to be practical [278]. Another advantage is its efficient transduction to enable the infecting of both dividing and non-dividing cells, which increases their applicability across different cell types [279]; neurotropism as HSV naturally infects neuronal cells, making it a candidate for gene therapies aimed at treating neurodegenerative diseases [280]. Unlike other vectors that integrate into the host genome and risk causing mutagenesis, HSV vectors typically remain episomes in the nucleus [281]. However, the cons of HSV can induce immune responses, which may limit their efficacy and make repeated administration problematic [282]. The large genome and complex life cycle of HSV make vector production and quality control more challenging compared to other viral vectors like AAV (adeno-associated virus) [283]. There are concerns about the potential reactivation of the viral genome, leading to herpetic disease. Thus, given the potential safety risks, numerous regulatory challenges must be overcome before HSV vectors can be used in clinical applications [284].

### 8.5. Direct

Several difficulties are involved in directly injecting nucleic acids (like plasmid DNA or mRNA encoding Cas9) for in vivo gene editing. Large polynucleotide molecules like DNA and RNA are hydrophobic, negatively charged, and unstable [285]. The cell membranes are attracted to these physiochemical characteristics, preventing them from entering cells independently. They also have a short half-life in circulation because serum nuclease activity makes it impossible for unprotected nucleic acids to reach specific target regions. The kidneys quickly remove these nucleic acids from the body and may stimulate the immune system by interacting with pattern recognition receptors. It calls for sophisticated packaging and delivery systems to overcome these obstacles.

However, these transfection reagents' cytotoxic and inflammatory effects, including lipofectamine, restrict their use in vivo applications. Many of these obstacles have been removed by creating novel synthetic ionizable cationic lipids and LNP formulations, opening the door to the possibility of LNP-mediated therapeutic gene editing.

### 8.6. LNPs

The arrival of COVID-19 mRNA vaccines brought many breakthroughs, including using lipid nanoparticles to deliver mRNA; it is now highly likely that more GE products will adopt this technology that has matured [286]. However, issues related to the excipients used in lipid nanoparticles, particularly polyethylene glycol, have recently been identified as a new source of anti-PEG antibodies [287] besides the reported skin reactions [288]. These findings would not have been possible had the vaccine not been administered to billions of subjects; it now requires a review of the safety of the LNP formulations developed for GE products [273].

LNPs are a popular therapeutic nucleic acid delivery method. Ionizable cationic lipids, polyethylene glycol (PEG) lipids, zwitterionic phospholipids, and cholesterol are their typical four main lipid constituents. In addition, most LNPs enter cells through the endocytosis pathway. These foundational elements endow LNP systems with special functional elements that cooperate to enable payload encapsulation, transport, and delivery.

LNPs' capacity to avoid recognition by the innate immune system and longer circulation time make them advantageous as drug carriers [289]. These characteristics are beneficial for delivering hydrophobic medications with brief circulation half-lives, such as proteins and nucleic acids. LNPs containing CRISPR components in nucleic acid or protein forms can effectively induce on-target therapeutic GE in target tissues when given enough circulation time. In addition, ready-to-use LNP formulations can be adopted with little effort.

### 8.7. Nano Particles

One tested method uses nanoparticles endocytosed by the target cell and functions in the nucleus or cytoplasm. Non-viral nanoparticles can deliver genome-editing tools in vitro, ex vivo, and in vivo, occasionally in conjunction with viral vectors. They are frequently made from synthetic, cationic lipid, or polymer delivery materials. CRISPR-Cas9 can also be delivered using a system that combines magnetic nanoparticles (MNPs) and recombinant baculoviral vectors (BVs).

Nanospheres and nanocapsules are two families of nanoparticles that can be distinguished structurally. Nanospheres contain a uniform matrix throughout the particles that store active substances, unlike nanocapsules with a core-shell structure with the payload inside the inner core. Lipid moieties are found in the structures of lipid-based nanoparticles, which have enormous biomedical potential for gene therapy and drug delivery. Lipid-based nanoparticles have several advantages over viral and nonviral nanoparticle systems, including simplicity in formulation, spontaneous self-assembly, high potency, high biocompatibility, a greater payload capacity, and adaptability in application design [290].

For example, after intravenous injection, PEGylated CLAN nanoparticles and LNPs could transfer plasmids or RNA-based CRISPR-Cas into different immune cells or hepatocytes. It is essential to carefully research the in vivo destiny of these systemically administered nanoparticles. Additionally, it is critical to evaluate the safety risks associated with generating GE in healthy cells like hematopoietic stem cells and germ cells. Many non-target cells may endocytose these systemically injected nanoparticles [291]. However, a single dose of nanoparticles can produce effective and long-lasting editing [292].

Some delivery systems for systemic injection can inject drugs locally to trigger local GE into a tumor, muscle, skin, eye, ear, or brain. For instance, DNA nano clews could deliver the CRISPR-Cas RNP after intertumoral injection to edit tumor genes [293]. After intramuscular injection, AuNPs could also have donor DNA and CRISPR-Cas RNPs to fix the mutated dystrophin gene [190]. Tailoring charge, hydrophilicity, and functional ligands in AuNPs are simple [294].

Although local GE only has a small number of target cells, this approach has few security concerns. Many vectors used in gene therapy, which typically emphasizes long-term expression to correct genetic flaws, are rarely appropriate for GE, which only needs temporary delivery of editors. The most popular editors also present additional challenges because of their large sizes (SpyCas9 and TALENs), repetitive sequences, and need to deliver both components of a ribonucleoprotein complex (ZFNs and TALENs), as well as their large sizes (SpyCas9 and TALENs) (RNP; for example, in CRISPR). Accurate tissue targeting is also necessary due to the possibility of on-target or off-target activity in the incorrect tissues [270].

### 8.8. Plasmid

The broad CRISPR-Cas system leaves plasmids with a significant negative charge. As a result, CRISPR-Cas systems must be contained within delivery systems or constrained and compressed into small sizes. Furthermore, delivery techniques that can get beyond in vivo barriers, such as nuclease degradation, immunogenicity, specific cell targeting, and nuclear envelope barriers, might significantly increase GE effectiveness and lessen the possibility of off-target effects. The capacity of GE can also be increased by delivery devices that simultaneously contain ssDNA or dsDNA. More donors of single-strand DNA (ssDNA) or double-strand DNA (dsDNA) are required for gene insertion or repair. Cas mRNA and gRNA are less harmful than plasmids, which carry the risk of genome integration [295]. Cas mRNA has a short half-life and can degrade in as little as 24 h, which lowers the likelihood of immunogenicity and off-target effects. Using dsDNA templates with sequences derived from pCas and transcription driven by the T7 promoter, Cas mRNA and gRNA are typically produced in vitro. Any CRISPR-Cas on a plasmid can be translated into RNA and delivered. The length of the Cas mRNA varies from 3000 nucleotides (nt) to over 5000 nt because of the different types of Cas proteins and their modifiers.

HEK293FT, U2OS, murine ESCs, N2A, and A549 cell lines can be transfected with plasmid-based Cas9/gRNA, RNA mixes of Cas9 and sgRNA, and even RNPs using commercially available transfection reagents designed for plasmid and siRNA delivery. Exosomes can be modified to deliver RNA-based CRISPR-Cas in addition to plasmids.

### 8.9. Ribonucleoproteins (RNP)

RNPs are made up of a big Cas protein and a short gRNA. gRNA can bind to DNA by Watson-Crick base pairing, or the Cas protein can couple to polypeptides, proteins, and PEI. These characteristics allow for the loading of RNP as well. RNP can also be loaded by electrostatic interactions with positively charged objects because of its negative net charge. Metal-organic frameworks (MOFs), polypeptides, PEI, cationic lipids, and others are among these positively charged compounds. RNP can also be delivered through cell vesicles. Since RNPs have a negative net charge, they can be directly transfected without cationic liposomes or LNPs. Positively charged PEI has also been produced for the delivery of RNPs. Cas9 has been conjugated with proteins or polypeptides to make RNP administration easier. In addition to conjugation, it is also possible to complex or encapsulate the RNP with polypeptides. For the delivery of Cas9 RNPs, metal-organic frameworks (MOFs) have been created [296].

## 9. Prospective Views

Gene editing, particularly technologies like CRISPR/Cas9, has revolutionized the field of genetics and holds immense potential for various applications, from medical treatments to agricultural advancements. How fast these developments occur will depend on the human risk, development cost, and social motivation. The table classifies this prospective possibility into four categories (Table 4).

**Table 4.** Prospective Analysis of GE.

| Field | Future Developments |
|---|---|
| Fastest Development | |
| Agriculture | The GMOs are already on the table. GE will further revolutionize agriculture by creating more nutritious crops resistant to pests and diseases or adaptable to changing climate conditions [297]. Creating disease-resistant crops [180,298] that bring greater food security [299]. |
| Biocomputing | Using DNA in place of silicon chips is already in the early stages of development for data storage and processing [300]. Computers will be built inside living cells to perform complex computations [301]. |
| NGS | The next-generation sequencing (NGS) technologies will expand fast and become available at the point of use in clinics, allowing more relevant and critical diagnosis, designing personalized therapies, and preventing long-term illnesses. The cost of this diagnosis will be fully reimbursed since it will prove to be the most significant measure in reducing the cost of healthcare. |
| Bioenergy and Environment | Algae and certain bacteria can be edited to produce biofuels more efficiently, and gene editing can also help in bioremediation, where organisms are modified to clean up environmental pollutants [302,303]. Industrial yeasts can produce biofuels and various other chemicals; GE can improve the efficiency and versatility of these yeasts [304]. Gene editing can produce more efficient biofuels or bioplastics by altering microbial pathways, contributing to a more sustainable future [305]. |
| Biomaterials | Create organisms that produce new biomaterials with unique properties, opening various industrial and scientific applications [306]. |
| Biosensors | Engineer cells to detect specific molecules or conditions, creating biosensors for various applications, from medical diagnostics to environmental monitoring [307]. |
| Conservation Efforts | Gene editing could aid conservation by introducing genetic diversity into endangered populations or by engineering invasive species to limit their reproduction [308]. By creating precise genetic modifications, scientists can study evolutionary pathways, development processes, and the intricate interactions of genes in real time [309]. |

**Table 4.** *Cont.*

| Field | Future Developments |
|---|---|
| CRISPR/Cas9 | CRISPR/Cas 9 has revolutionized the field of genetics and holds immense potential for various applications, from medical treatments to agricultural advancements.<br>Beyond editing, the CRISPR system will be adapted for diagnostics. Platforms like SHERLOCK (Specific High-sensitivity Enzymatic Reporter UnLOCKing) can detect specific DNA or RNA sequences in pathogens [310]. |
| Environment | Editing the genes of certain bacteria to make them produce biodegradable plastics offering an environmentally friendly alternative to conventional plastics [311].<br>To bolster conservation efforts by creating white-footed mice immune to the bacteria causing Lyme disease [308]. Gene drives using CRISPR/Cas9 systems have been proposed to control disease vectors, such as mosquitoes that spread malaria [312]. |
| Nutrition | Gene editing can enhance the nutritional content of food crops, potentially addressing malnutrition problems in areas of the world where specific nutrient deficiencies are common [313].<br>To create versions of common foods that do not trigger allergic reactions. For example, researchers have used gene editing to create a variety of wheat that does not produce the proteins that cause most wheat allergies [314]. |
| Personalized Medicine | With an understanding of individual genetic makeup, treatments can be tailored specifically to the genetic profile of each patient, offering better outcomes [315]. |
| Synthetic Biology and Biomanufacturing | New organisms can be designed to produce complex organic compounds, pharmaceuticals, or even materials for manufacturing [316]. |
| Vaccination | Creating attenuated strains of pathogens for more effective vaccines [317]. |
| *Slower Development* | |
| Drug Development | Creating models of human diseases in animals, providing a platform to test new drugs more efficiently using modeling that is not currently available [5]. |
| Modifying Microbiomes | The collection of microbes living in and on us, the microbiome, plays a crucial role in our health; they can be modified to promote health and combat diseases [318]. |
| Organ Transplants | Using animals like pigs, gene editing can help produce organs suitable for human transplantation [107]. Modify donor organs to increase compatibility with recipients, potentially reducing organ rejection rates [319]. |
| *Slowest Development* | |
| Animal Welfare | Improve the well-being of animals; for instance, pigs can be edited to be resistant to diseases, or chickens can be edited to only produce female offspring for egg production [320].<br>Researchers can use gene editing to create cell cultures or organoids that mimic human tissues for drug testing and disease modeling as an alternative to live animal testing [321]. |
| Gene Therapeutics | Targeting and repairing the faulty genes by GE will likely eradicate diseases like cystic fibrosis, Duchenne muscular dystrophy, and sickle cell anemia, which are prime candidates [322].<br>Beyond editing DNA sequences, tools are being developed to modify the epigenome, which could offer treatments for diseases where epigenetic changes play a role [323].<br>Many rare genetic disorders, often neglected by mainstream research due to their low prevalence, could be targeted and potentially cured using gene editing [16]. |
| Infectious Diseases | To bring extinct species back to life, or "de-extinction". This would involve using DNA from preserved specimens to edit the genes of a closely related existing species [324].<br>While controversial, gene editing technologies like CRISPR have raised the possibility of enhancing human abilities beyond normal levels, or "human enhancement" [325]. |
| Life Augmentation | By targeting genes involved in aging processes, gene editing may offer strategies to extend healthy human lifespan or combat age-related diseases [326].<br>For conditions like congenital blindness or deafness due to specific gene mutations, gene editing offers a path to restore these senses [327].<br>There is potential (albeit controversial) for gene editing to enhance human capabilities, such as increased strength, improved cognitive function, or resistance to diseases [328].<br>The possibility of enhancing cognitive abilities through gene editing is being explored. While this poses vast ethical dilemmas, it could revolutionize cognitive disorders or brain injury treatments [329]. |
| Neurodegenerative Diseases | Gene editing offers promise in the treatment or delay of neurodegenerative diseases like Alzheimer's, Parkinson's, and Huntington's by targeting the causative genes or modifying disease pathways [65]. |
| *Least Likely* | |
| Species Modulation | "Designer babies" (e.g., in China in 2018) and species modulations are highly possible since modifications can be passed on to subsequent generations [195]. There is a global consensus to prevent this, but as history will tell, it is impossible to contain a knowledge base. This will happen sooner or later, which may be the pivotal step in human evolution; it is of little concern to the process, whether it is brought in by long-time mutations or planned. We may be responsible for our evolution, not as an accident but as a random evolution design. |

## 10. Conclusions

GE will change the map of human destiny when it arrives and if it becomes accessible to billions who need it most. Unlike a biological drug, it has no pharmacology or contact toxicity pattern. The RNA-based CRISPR tool is well-characterized and specific to its target. Unlike biological drugs, there is little variability from batch to batch. It is almost like chemical drugs since the structures of its components are well-defined. With expanding PCR technology, producing a cell-free RNA that goes with Cas9 will be possible. The likely caution is the possibility of off-target genome editing, and the tests created to measure this have proven unreliable. Additionally, as these tools can only be tested in patients, there is a need for regulatory agencies to allow faster testing once GMP compliance is in order. Given the available GMP-grade materials, all off-the-shelf, this should lead to the development of many products without spending billions of dollars.

The future of GE tools will depend on bringing more rationality at the regulatory level and more creativity at the development stage to avoid facing the price affordability constraints that are now holding back gene therapies. The humanitarian cause becomes more relevant as we advance in lifesaving rare disease management; the cost of goods of these products is minuscule, yet the intellectual property protections let these products demand pricing that is out of reach of most patients. The high development cost can be optimized by bringing more science into regulatory requirements; however, given the novel nature of GE products and the risk of off-target impact, it is not likely. However, once the intellectual property and exclusivities expire, the regulatory agencies should consider allowing entry of these products in a plan comparable to what is allowed for biological therapeutic proteins—as biosimilars. The GT and GE products do not qualify as therapeutic proteins, requiring the regulatory agencies to start planning for approval pathways for biologically similar GE and GT products. The FDA has taken a major step by issuing a regulatory guideline specific to GE recently; while there is still a need for much clarification and explanation for the requirements suggested by the FDA, we will have to wait till the FDA begins approving these products, to understand the FDA perspective better. The FDA is anticipating approving many of these products. This event will revolutionize healthcare as we enter an era of managing diseases without treatment. This prospective possibility is indeed a revolution, and we are significantly close to conquering the greatest sufferings of humanity.

**Funding:** This research received no external funding.

**Conflicts of Interest:** The author declares that the research was conducted without any commercial or financial relationships that could be construed as a potential conflict of interest.

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
