# Peer review of "The Dawn of In Vivo Gene Editing Era: A Revolution in the Making"

_biologics, doi:10.3390/biologics3040014_

Round 1
Reviewer 1 Report
The review article by Dr. Sarfaraz K. Niazi on the current status of gene editing is fairly compressive as well as timely and important, albeit somewhat disjointed. There are no substantive criticisms, but the author should consider the following points, which would be helpful to the reader.
1. Since the article also encompasses gene therapy (GT), it would make sense to modify the title as well as the abstract and the rest of the manuscript, where appropriate, to reflect this fact.
2. The manuscript might read better if it is sub-divided into two sections – gene therapy and gene editing.
3. Some of the serious adverse events, including patient deaths, in gene therapy trials with lentiviral vectors (see Nature, 467: 318-322, 2010; N Engl J Med, 386: 138-147, 2022); and AAV vectors (see Mol Ther Nucl Acids, 32: 949-959, 2023, should be mentioned, and these references cited.
4. The potential concerns regarding long-term safety of CRISPR/Cas9 should also be mentioned (see Mol Ther Meth Clin Dev, 23: 507-523, 2021; bioRxiv, 2023 Mar 22:2023.03.22.533709. doi: 10.1101/2023.03.22.533709), and these references cited as well.
Reviewer 2 Report
Sarfaraz K. Niazi’s review is a much-needed overview of the gene editing therapeutics field and was insightful and comprehensive. I feel the article could be improved in the flow and interpretation. I have the following recommendations:
[Line 47] It remains to be seen how useful gene editing will be in treating a genetically heterogenous group of diseases like cancers. Therefore, it seems overly simplistic for the author to state “all cancers” in a list of single-gene disease. The application of gene editing to cancer therapeutics in itself is novel and particularly interesting considering the many hallmarks of cancer and the occurrence of therapy escape. This topic may warrant more discussion and attention from the author than what is currently given in the article.
With regards to off-target surveillance, genetic diversity is an important consideration when designing and evaluating gene editing products (e.g. https://doi.org/10.1038/s41588-022-01257-y) yet it is not mentioned within the article.
[Lines 121-123] It would be insightful to additionally illustrate how many gene editing clinical trials are underway at different health authorities and at what phases (e.g. FDA, MHRA, New Zealand etc). Enhanced by a comment on the regulation by health authority it would be particularly insightful in discerning the international landscape/appetite for gene editing treatment.
The article may flow better if the “Challenges and Concerns” section comes after the “Current Status” section and the author should add a “Future Progress” or similar section to allow the reader to understand the progress and technologies needed to push the field forward.
[Line 70-71] While next generation sequencing has made individual therapies practical, it is unclear how widespread NGS-based testing is in routine medical care and whether its use is enough to allow patients to be matched with potential therapies. The author could comment more on this and on the synergy needed between the two technologies (perhaps in future progress section). Additionally, if NGS tests are covered by health insurance, for particular genetic susceptibilities (e.g. cardiovascular disease), would the natural extension be to cover the gene editing therapy for variants identified to provide clinical utility of the test?
The author could comment on whether the therapeutic effect seen in preclinical studies is likely to translate to therapeutic effect in humans. For example in clinical trials, not all patients achieve a response and sometimes the therapeutic response is moderate in comparison to that observed in non-human primates (e.g. LCA trial EDIT-101 and recent article on Novartis gene editing drug Zolgensma https://www.reuters.com/business/healthcare-pharmaceuticals/what-happens-when-2-million-gene-therapy-is-not-enough-2023-08-12/).
Many tables in the article are formatted like lists and are not particularly useful or easy to digest in this format. For example, it may be more useful for Table 1 to by stratified by application by containing an additional column which could group similar rows, e.g. Column 1 row “Agriculture” and then column 2 contains all use-cases for Agriculture in sequence.
[Line 13-15] The statement that gene editing concerns will be resolved upon FDA approval of GE therapeutics is not yet known. Perhaps the author meant to use a word similar to clarified.
[Lines 85-87] The table is annotated as Table 3 but should be Table 2.
[Line 124] Table 2 should be annotated as Table 3.
[Line 256-258] would be better suited in the section outlining ethical considerations.
[Line 392] Sentence seems to end abruptly.
[Line 34-35] Could not locate the reference listed (ref. 2). It must be noted that this reference is quite old, but neither a doi nor title search brought it up.
[Line 29] A reference for CRISPR cas9 discovery would be useful here.
Reviewer 3 Report
The review manuscript by Niazi S. K., entitled “In Vivo Gene Editing: A Healthcare Revolution in the Making” comprehensively summarized the applications of in vivo gene editing. I appreciate the author for his commendable efforts.
However, there are few minor issues (listed below) which need to be addressed with appropriate corrections. Over all, this manuscript is recommended for publication after the revision.
Minor Concerns:
1. In AAV section, using AAV for gene editing is confusing. However author mentioned in the lines 433-439, it is important to mention the serotype. Only AAV6 is used for the application so far. Other AAV serotypes are being used in supplying donor template for HDR editing.
2. Among the in vivo gene editing techniques for human applications, base editing technology is found to be safer and has large potential applications. Multiple clinical trials are being pursued. The author should include those details for making the review more effective.
Reviewer 4 Report
In this review by Sarfaraz K. Niazi, the author did an elaborate literature review for gene editing and gene therapy in the field, shedding light on the importance of treating untreatable diseases. Nonetheless, there are a few issues that need to be addressed.
Comments:
· The author's comprehensive literature review is commendable. The clear categorization of strategies for treating mutations that cause hereditary diseases into Gene Editing (GE) and Gene Therapy (GT) is a positive aspect. However, there are instances where the content appears to deviate from the manuscript's title, "In Vivo Gene Editing: A Healthcare Revolution in the Making."
· The total flow of this review needs to be improved. Too much redundancy makes the manuscript sporadic. In addition, I would suggest introducing the gene editing tools prior to discussing their applications, along with related discussions and concerns.
· There are two sections labeled "Current Status," containing numerous duplicated and redundant contents that need consolidation.
· Personally, I believe that incorporating original insights from the author is preferable to solely presenting a figure (Figure 1) or a table (Table 5) without added contextual value.
· Several concepts are introduced abruptly, such as anti-CRISPR and DIY kits.
· Ensure that when abbreviating terms, the full term is spelled out the first time it's used. Avoid repetitive use of the full term after abbreviation. Terms like GMO, ATMP, PMDA, SSNs, Cas9n, and PEI need to be fully named first. Please double-check throughout the manuscript.
· In table 1, 3, please use superscript for reference numbers.
· Line 47, It's inaccurate to classify all cancers as single-gene diseases.
· Line 69, “the approval of 17 personalized (individualized, precision) medicines”. Approved by what?
· In table 3, some clauses are overlapped with each other. For example, Duchenne Muscular Dystrophy & Muscular Dystrophy, Restoring Sight & Retinal Diseases…
· Ensure that table 2 is positioned before table 3.
· Could you please update the FDA-approve Gene Therapy products before you finalize this manuscript?
· Line 94 and 179, the birth of gene-edited babies in China was in 2018 or 2019?
· Line 208-210, “employ T cells modified by gene editing to resist HIV infection”, is not a case of "excise the integrated HIV genome from infected cells in vitro". They are two different methods.
· Figure 3, inversion was not shown in panel B.
· Line 367, “reducing on- and off-target effects”, why would we reduce on-target effects?
· In “primer editing”, “its capacity to edit farther from the Cas9 nicking site, it promises greater targeting flexibility”, this is not specific to primer editing, it's for Cpf-1. However, Cpf-1 is not a necessity in primer editing.
· Line 413-421, the concept for Homologous Recombination described in the manuscript was too limited and narrowed
· Line 423, AAV is widely known as a gene delivery tool, so here the author should have given more information to claim it as a gene editing tool.
· what is the editing efficiency when using AAV as a gene editing platform? what are the pros and cons?
· It doesn't make sense for the description of ssODNS as a GE tool in the manuscript. It seems that the author just excerpted from this paper. https://pubmed.ncbi.nlm.nih.gov/30886178/
· It might be irrelevant to include synthetic genomics in this manuscript.
· Line 536-539, “as well as their large sizes (SpyCas9 and TALENs)” is repetitive.
· Line 542, Is there any evidence for AAV vectors to be tumorigenic? This is a serious side effect.
· Line 551, What’s the full name of HSV? The pros of HSV were listed but what are the cons to make it not the first choice for viral gene delivery?
· Most of the “delivery tools” that the author described are not related to in vivo therapy. It's kind of beside the point.
Reviewer 5 Report
Having thoroughly reviewed the article written by Dr. Niazi, I am pleased to express my positive view on its content and quality. I believe that the review article makes a significant contribution to the field of gene and genome editing, shedding light on the critical distinctions between gene editing (GE) and gene therapy (GT), while also discussing the potential of these technologies in treating various diseases. The distinction between GE as a tool and GT as a form of medicine is eloquently outlined. The identification of the current regulatory landscape, with 22 FDA-approved GT products in contrast to the yet-to-be-approved in vivo GE products, highlights the evolving nature of this field and its potential for rapid advancements. The author adeptly emphasizes the immense therapeutic potential of GE in treating a wide array of diseases. The recognition of the pivotal role genes play in human diseases underscores the significance of GE as a potential therapeutic avenue. Additionally, I appreciate the author's acknowledgement of the inherent challenges in GE, including off-target effects, delivery consistency, and long-term implications, which are crucial considerations in the responsible development of these technologies.
Furthermore, I believe the author could consider to add a small section highlighting about the recent advances in mitochondrial genome engineering in vivo and its potential, which is indeed relevant and adds depth to the article. I will highly recommend going through a recent review article emphasizing this topic (https://www.mdpi.com/1422-0067/24/6/5798). The author could add a paragraph to this manuscript highlighting this recent and exciting development and its potential.
Considering the valuable insights presented in the review article, I wholeheartedly recommend its publication.
Author Response
THANKS FOR YOUR DILIGENCE IN REVIEWING AND SUGGESTING VALUABLE CHANGES
Furthermore, I believe the author could consider to add a small section highlighting about the recent advances in mitochondrial genome engineering in vivo and its potential, which is indeed relevant and adds depth to the article. I will highly recommend going through a recent review article emphasizing this topic (https://www.mdpi.com/1422-0067/24/6/5798). The author could add a paragraph to this manuscript highlighting this recent and exciting development and its potential. FULL SECTION ON MITOCHODRIAL GENOME IS ADDED